# $\mathcal{O}$-GNN: INCORPORATING RING PRIORS INTO MOLECULAR MODELING[*]

[1]**Jinhua Zhu**, [1]**Kehan Wu**, [1]**Bohan Wang**, [2]**Yingce Xia**, [3]**Shufang Xie**, [2]**Qi Meng**,
[2]**Lijun Wu**, [2]**Tao Qin**, [1]**Wengang Zhou**, [1]**Houqiang Li**, [2]**Tie-Yan Liu**
[1]University of Science and Technology of China, [2]Microsoft Research AI4Science
[3]Gaoling School of Artificial Intelligence, Renmin University of China
[1]{teslazhu, wu_2018}@mail.ustc.edu.cn, [1]bhwangfy@gmail.com,
[1]{zhwg,lihq}@ustc.edu.cn, [3]shufangxie@ruc.edu.cn,
[2]{yingce.xia, meq, lijuwu, taoqin, tyliu}@microsoft.com

## ABSTRACT

Cyclic compounds that contain at least one ring play an important role in drug design. Despite the recent success of molecular modeling with graph neural networks (GNNs), few models explicitly take rings in compounds into consideration, consequently limiting the expressiveness of the models. In this work, we design a new variant of GNN, ring-enhanced GNN ($\mathcal{O}$-GNN), that explicitly models rings in addition to atoms and bonds in compounds. In $\mathcal{O}$-GNN, each ring is represented by a latent vector, which contributes to and is iteratively updated by atom and bond representations. Theoretical analysis shows that $\mathcal{O}$-GNN is able to distinguish two isomorphic subgraphs lying on different rings using only one layer while conventional graph convolutional neural networks require multiple layers to distinguish, demonstrating that $\mathcal{O}$-GNN is more expressive. Through experiments, $\mathcal{O}$-GNN shows good performance on **11** public datasets. In particular, it achieves state-of-the-art validation result on the PCQM4Mv1 benchmark (outperforming the previous KDDCup champion solution) and the drug-drug interaction prediction task on DrugBank. Furthermore, $\mathcal{O}$-GNN outperforms strong baselines (without modeling rings) on the molecular property prediction and retrosynthesis prediction tasks. The code is released at `https://github.com/O-GNN/O-GNN`.

## 1 INTRODUCTION

Cyclic compounds, which refers to the molecules that have at least one ring in its system, naturally exist in the chemical space. According to our statistics on $109M$ compounds from PubChem (Kim et al., 2019) which is a widely used chemical library, more than $90\%$ compounds have at least one ring. The rings could be small/simple (*e.g.*, the benzene is a six-member carbon ring, and the pentazole is a five-member nitrogen ring) or large/complex (*e.g.*, the molecule shown in Figure 1). Rings are important in drug discovery, for example: (1) Rings can potentially reduce the flexibility of molecules, reduce the uncertainty when interacting with target proteins, and lock the molecules to their bioactive conformation (Sun et al., 2012). (2) Macrocyclic compounds, which usually have a ring with more than 12 atoms, play important roles in antibotics design (Venugopal & Johnson, 2011) and peptide drug design (Bhardwaj et al., 2022).

Recently, deep neural networks, especially graph neural networks (denoted as GNN) (Kipf & Welling, 2017; Hamilton et al., 2017a), have been widely used in molecular modeling. A GNN takes a graph as input, and messages of different nodes are passed along edges. GNNs have made great success in scientific discovery: (1) Stokes et al. (2020) train a GNN to predict growth inhibition of Escherichia coli and find that Halicin is a broad-spectrum bactericidal antibiotic. (2) Shan et al. (2022) leverage GNN to model the interactions between proteins, and they eventually obtain possible antibodies for SARS-CoV-2. In addition, GNNs are widely used in drug property prediction (Rong et al., 2020), drug-target interaction modeling (Torng & Altman, 2019), retrosynthesis

---

[*]This work was done when Jinhua Zhu, Kekan Wu and Bohan Wang were interns at Microsoft Research AI4Science. Correspondence to: Yingce Xia.

(Chen & Jung, 2021), etc. However, none of the above work explicitly models the ring information into GNNs. From the application's perspective, they miss an important feature for their tasks. From the machine learning's perspective, Loukas (2020) points out that existing message-passing-based GNNs cannot properly capture the ring information when the product of network width and height is not large enough (see the Table 1 in Loukas (2020)). Therefore, with the classic GNNs, the ring information in compounds is not well leveraged.

To tackle this issue, in this work, we propose a new model, ring-enhanced GNN (denoted as $\mathcal{O}$-GNN), that explicitly models the ring information in a compound. The $\mathcal{O}$ stands for the rings in molecules and is pronounced as "O". Generally speaking, $\mathcal{O}$-GNN stacks $L$ layers, and each layer sequentially updates edge representations, node representations and ring representations by aggregating their neighbourhood information. We mainly use a self-attention layer for adaptive message passing, and use a feed-forward layer to introduce non-linearity to representations.

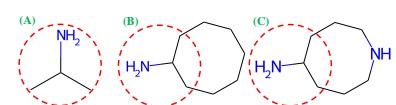

Figure 1: Paclitaxel, a compound with 7 simple rings. Kampan et al. (2015) summarized that the intact taxane ring (i.e., $r_4$, $r_5$, $r_6$) and a four-membered oxetane side ring (i.e., $r_7$) is essential to induce cytotoxic activity.

We first demonstrate the advantage of $\mathcal{O}$-GNN through theoretical analysis. $\mathcal{O}$-GNN is able to distinguish two isomorphic sub-graphs lying on different rings using only one layer (see Figure 2 for the example). On the contrary, if we remove the ring-modeling components from $\mathcal{O}$-GNN, such a distinguishability would require multiple layers (see Section 2.3 for detailed analysis). These results demonstrate that $\mathcal{O}$-GNN is more expressive than conventional graph convolutional networks in the absence of explicitly modeling rings.

We then conduct experiments on 11 datasets from three tasks, including molecular property prediction, drug-drug interaction prediction and retrosynthesis:

(1) For molecular property prediction, we first conduct experiments on PCQM4Mv1, which is to predict the HOMO-LUMO gap of molecules. Our method outperforms the champion solution of KDDCup on the validation set (Shi et al., 2022) (note that test set labels are not available). Next, we verify $\mathcal{O}$-GNN on six datasets from MoleculeNet (Wu et al., 2018), which is to predict several pharmaceutical related properties of molecules. $\mathcal{O}$-GNN outperforms the corresponding GNN baselines without rings. Finally, we conduct experiments on FS-Mol (Stanley et al., 2021), a few-shot property prediction task, and shows that modeling rings can also improve the prediction accuracy. (2) For drug-drug interaction prediction, which is to predict whether two drugs interacts with each

Figure 2: An illustrative example of theoretical results. The three substructures in the red circles are isomorphic. The second and third substructures lie on different rings (a Cyclooctane and an Azocane). A regular GNN requires multiple layers to distinguish the three substructures while $\mathcal{O}$-GNN requires only one layer due to the ring representations.

other, we test $\mathcal{O}$-GNN on DrugBank following the previous settings (Nyamabo et al., 2021; Li et al., 2022), and achieve state-of-the-art results. (3) For retrosynthesis, we apply $\mathcal{O}$-GNN to LocalRetro (Chen & Jung, 2021), a strong GNN-based method for retrosynthesis. On USPTO-50k, our method significantly boosts the accuracy.

## 2 METHOD

### 2.1 NOTATION AND PRELIMINARIES

Let $G = (V, E)$ denote a molecular graph, where $V$ and $E$ are the collections of nodes/atoms and edges/bonds[1]. Let $R$ denote the collection of rings in $G$. Define $V = \{v_1, v_2, \cdots, v_{|V|}\}$ and

---

[1]When the context is clear, we use nodes/atoms and edges/bonds alternatively in this work.

$E = \{e_{ij}\}$, where $v_i$ is the $i$-th atom and $e_{ij}$ is the bond connecting $v_i$ and $v_j$. When the context is clear, we use $i$ to denote atom $v_i$, and use $e(v_i, v_j)$ to denote edge $e_{ij}$. Let $\mathcal{N}(i)$ denote the neighbors of atom $i$, i.e., $\mathcal{N}(i) = \{v_j \mid e_{ij} \in E\}$. Define $R = \{r_1, r_2, \cdots, r_{|R|}\}$, where each $r_i$ is a simple ring. A simple ring does not contain any ring structure. For example, for the molecule in Figure 3, it has two simple rings as we marked ($r_1$ and $r_2$). The ring $(1, 2, 3, 4, 5, 6, 7, 8, 9, 1)$ is not a simple ring. Let $R(v_i)$ and $R(e_{ij})$ denote the rings that the atom $v_i$ or the bond $e_{ij}$ lies on, and $V(r)$ and $E(r)$ denote all the atoms and the bonds lying on ring $r$. For example, in Figure 3, $R(v_4) = \{r_1, r_2\}$ while $R(v_3) = r_2$. $R(e_{49}) = \{r_1, r_2\}$ while $R(e_{78}) = r_1$. $V(r_1) = \{v_4, v_5, v_6, v_7, v_8, v_9\}$ and $E(r_1) = \{e_{45}, e_{56}, e_{67}, e_{78}, e_{89}, e_{94}\}$.

A graph neural network (GNN) is usually stacked by several identical GNN layers. Each GNN layer is composed of an Aggregate function and an Update function,

$$h_i' = \text{Update}\left(h_i, \text{Aggregate}(h_j | j \in \mathcal{N}(i))\right), \tag{1}$$

where $h_i$ is the representation of atom $i$ and $h_i'$ is its updated representation. Different GNNs have different Aggregate functions and Update functions. Details are summarized in Appendix D.

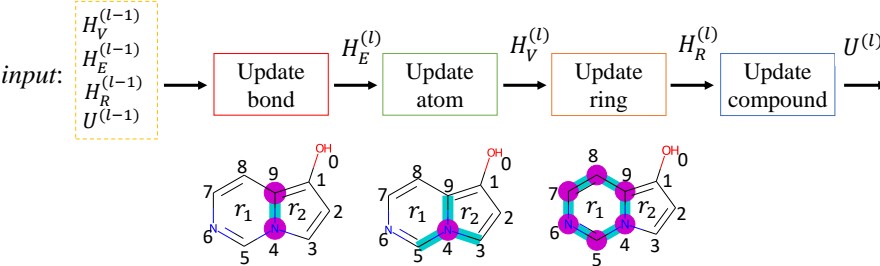

Figure 3: The workflow of our method. $H_E^{(l)}$, $H_V^{(l)}$, $H_R^{(l)}$ and $U^{(l)}$ denote the representation collections of bond, atom, ring and the global compound at the $l$-th layer.

## 2.2 MODEL

Our model consists of $L$ identical layers with different parameters. The architecture of each layer is shown in Figure 3. Let $h_i^{(l)}$, $h_{ij}^{(l)}$ and $h_r^{(l)}$ denote the output representations of atom $v_i$, bond $e_{ij}$ and ring $r$ at the $l$-th layer, respectively. Let $U^{(l)}$ denote the compound representation at the $l$-th layer. We initialize $h_i^{(0)}$ via a learnable embedding layer which indicates its atomic type, chirality, degree number, formal charge, hybridization type, and so on. Similarly, we initialize $h_{ij}^{(0)}$ with a learnable embedding wich indicates its bond type, stereoisomerism type and whether the bond is conjugated. Then we initialize $h_r^{(0)}$ by concatenating the node and edge embedding and then transform it with a non-linear layer. Last, we initialize the compound representation with a learnable embedding. In each layer, we update representations of nodes, bonds, rings and the compound sequentially. We will frequently use $\text{MLP}(\cdots)$, a multi-layer perception network with one hidden layer, to build our model. The inputs of $\text{MLP}$ are concatenated as a long vector and processed by the network.

(1) *Update bond representations*: The representation of a bond is updated via the connected atoms, the rings that the bond belongs to and the compound representation from the last layer:

$$h_{ij}^{(l)} = h_{ij}^{(l-1)} + \text{MLP}\left(h_i^{(l-1)}, h_j^{(l-1)}, h_{ij}^{(l-1)}, \frac{\sum_{r \in R(e_{ij})} h_r^{(l-1)}}{|R(e_{ij})|}, U^{(l-1)}\right). \tag{2}$$

(2) *Update atom representations*: We use an attention model to adaptively aggregate bond representations into the centralized atoms. Mathematically,

$$\bar{h}_i^{(l)} = \sum_{j \in \mathcal{N}(i)} \alpha_j W_v \text{concat}(h_{ij}^{(l)}, h_j^{(l-1)});$$

$$\alpha_j \propto \exp(\boldsymbol{a}^\top \text{LeakyReLU}(W_q h_i^{(l-1)} + W_k \text{concat}(h_j^{(l-1)}, h_{ij}^{(l)}))); \tag{3}$$

$$h_i^{(l)} = h_i^{(l-1)} + \text{MLP}\left(h_i^{(l-1)}, \bar{h}_i^{(l)}, \frac{1}{|R(v_i)|} \sum_{r \in R(v_i)} h_r^{(l-1)}, U^{(l-1)}\right).$$

In Eqn.(3), the $W$'s are the parameters to be learned, and `concat` denotes concatenating the input vectors as a long one.

(3) *Update ring representations*: The ring representations are updated using `MLP` networks:

$$h_r^{(l)} = h_r^{(l-1)} + \texttt{MLP}\Big(h_r^{(l-1)}, \sum_{v_i \in V(r)} h_i^{(l)}, \sum_{e_{ij} \in E(r)} h_{ij}^{(l)}, U^{(l-1)}\Big) \tag{4}$$

(4) *Update the compound representation*:

$$U^{(l)} = U^{(l-1)} + \texttt{MLP}\left(\frac{1}{|V|}\sum_{i=1}^{|V|} h_i^{(l)}, \frac{1}{|E|}\sum_{i,j} h_{ij}^{(l)}, \frac{1}{|R|}\sum_{r \in R} h_r^{(l)}, U^{(l-1)}\right). \tag{5}$$

After stacking $L$ $\mathcal{O}$-GNN layers, we get the graph representation by a simple average pooling layer, i.e., $h_{\mathcal{G}} = \frac{1}{|V|}\sum_{i=1}^{|V|} h_i^{(L)}$, which could be utilized by graph classification tasks. For node classification tasks, we can add a classification head to $h_i^{(L)}$.

## 2.3 THEORETICAL ANALYSIS

In this section, we compare the distinguishability between standard GNN (without ring representations) and $\mathcal{O}$-GNN. In addition to the notations defined in Section 2.1, we define the valued version of a graph $G = (V, E)$ as a triplet $\text{VALUE}_f(G) = (V, E, f)$, where $f$ is a mapping storing feature information and mapping a node or an edge to its corresponding input feature (e.g., a 256-dimension representation). We call $f$ as a feature mapping on $G$.

**Definition 1** ($k$-neighbourhood node). *For a molecular graph $G = (V, E)$ and two nodes $u, v \in V$, we say $u$ is a $k$-neighbourhood of $v$ if there exists a path in $G$ connecting $u$ and $v$ with length no larger than $k$. More formally, $u$ is a $k$-neighbourhood of $v$ if and only if there exists a set of nodes $\{v_0, v_1, \cdots, v_t\} \subset V$, such that, $t \leq k$, $v_0 = v$, $v_t = u$, and for any $i \in \{0, \cdots, t-1\}$, $v_{i+1} \in N(v_i)$.*

We highlight here that $v$ is a 0-neighbourhood node (and thus a $k$-neighbourhood node with any $k \geq 0$) of itself.

**Definition 2** ($k$-neighbourhood sub-graph). *For a molecular graph $G = (V, E)$ and a node $v$ in $G$, we define the $k$-neighbourhood sub-graph of $v$ as the sub-graph composed of all $v$'s $k$-neighbourhood node. More formally, we slightly abuse the notations and denote the $k$-neighbourhood sub-graph of $v$ as $G(v, k) \triangleq (V(v, k), E(v, k))$, where*

$$V(v, k) \triangleq \{u \in V : u \text{ is a } k\text{-neighbourhood node of } v\},$$
$$E(v, k) \triangleq \{e(v_1, v_2) \in E : v_1, v_2 \in V(v, k)\}.$$

**Definition 3** (Equivalent valued graph). *For two valued graphs $\text{VALUE}_{f_1}(G_1) = (V_1, E_1, f_1)$ and $\text{VALUE}_{f_2}(G_2) = (V_2, E_2, f_2)$, we say that they are equivalent if (i). $G_1$ and $G_2$ are isomorphic, i.e., there exists a one-to-one mapping $\mathcal{P} : V_1 \to V_2$, such that the edges are preserved; (ii). $\mathcal{P}$ also preserves the value of edges and the the value of nodes, i.e., $\forall u, v \in G_1$,*

$$e(u, v) \in E_1 \Leftrightarrow e(\mathcal{P}(u), \mathcal{P}(v)) \in E_2,$$
$$f_1(u) = f_2(\mathcal{P}(u)), f_1(v) = f_2(\mathcal{P}(v)), f_1(e(u, v)) = f_2(e(\mathcal{P}(u), \mathcal{P}(v))).$$

With all the preparations above, we are now ready to define the graph feature extractor and its discriminatory ability.

**Definition 4** (Graph feature extractor and its discriminatory ability). *We say a mapping $\Phi$ is a graph feature extractor, if it maps a valued graph $\text{VALUE}_f(G)$ to a new feature mapping $\tilde{f}$ on $G$. We further allow $\Phi$ to be parameterized as $\Phi_\theta$, and call $\Phi_\theta$ a parameterized graph feature extractor.*

*For a parameterized graph feature extractor $\Phi_\theta$, we say $\Phi_\theta$ has the discriminatory ability for $k$-neighbourhood sub-graphs, if for any valued graphs $(G, f)$ and any two nodes $u, v$ in $G$, if the valued $k$-neighbourhood sub-graph of $u$ and $v$ (i.e., $(G(u, k), f)$ and $(G(v, k), f)$) are equivalent, there exists $\theta^\star$ such that $\Phi_{\theta^\star}((G, f))(u) \neq \Phi_{\theta^\star}((G, f))(v)$. In this case, we also say that $\Phi_{\theta^\star}$ can distinguish $u$ and $v$.*

We point out that $\{h_i^{(l)}\}_i \cup \{h_{i,j}^{(l)}\}_{i,j}$ defined by Eqn. (2, 3, 4, 5) is a parameterized feature extractor, and thus above provides a formal definition of $\mathcal{O}$-GNN's discriminatory ability.

The next proposition shows that without the ring representation, the $\mathcal{O}$-GNN needs at least $k+1$ layer to have the has the discriminatory power for $k$-neighbourhood sub-graphs.

**Proposition 1.** *Without the ring presentation, $\mathcal{O}$-GNN with no more than $k$ layers does not have the discriminatory ability for $k$-neighbourhood sub-graphs.*

Note that Proposition 1 can be easily extended to the conventional graph convolutional neural networks, which only aggregate information from 1-neighborhood nodes. We then show that with the ring representations, $\mathcal{O}$-GNN with only one layer has the discriminatory power.

**Proposition 2.** *If $u$ and $v$ lie on different rings, $\mathcal{O}$-GNN with only one layer can distinguish them.*

The proofs are deferred to Appendix B due to space limitation. From Proposition 1 and 2, we can see that $\mathcal{O}$-GNN is more expressive than the regular GNN that does not model rings. The regular GNN requires at least $k$ layers to distinguish two isomorphic $k$-neighborhood sub-graphs on different rings, while $\mathcal{O}$-GNN only requires one layer for this purpose (see the example in Figure 2). Comparing $\mathcal{O}$-GNN to a regular GNN with the same number of layers, modeling ring presentations constantly increases the percentages of parameters (irrelevant to $k$). However, a regular GNN may require $k$ layers to achieve the discriminatory power for $k$-neighborhood sub-graphs. When $k$ is large, $\mathcal{O}$-GNN will be much more parameter efficient. More discussions are in Appendix C.5.

## 3 EXPERIMENTS

To validate the effectiveness of our method, we test $\mathcal{O}$-GNN on the following three tasks: molecular property prediction, drug-drug interaction prediction and retrosynthesis. The first two tasks are graph classification tasks, and the third one is a node/link prediction task.

### 3.1 APPLICATION TO MOLECULAR PROPERTY PREDICTION

**Datasets.** We work on three datasets for this application:

(1) The HOMO-LUMO energy gap prediction of the PCQM4Mv1 dataset (Hu et al., 2021). The input is a 2D molecular graph, and the target is its HOMO-LUMO energy gap, which is an essential molecular property in quantum chemistry. PCQM4Mv1 has 3045360 and 380670 training and validation data (test labels are not available). The properties are obtained via density function theory.

(2) Molecular property prediction on MoleculeNet dataset (Wu et al., 2018). This is a dataset about the prediction of pharmaceutical properties of small molecules. We choose six molecular property prediction tasks (including BBBP, Tox21, ClinTox, HIV, BACE and SIDER), and the data ranges from 1.5k to 41k.

(3) Few-shot molecular property prediction of the FS-Mol dataset (Stanley et al., 2021). FS-Mol is made up of 5120 separate assays extracted from ChEMBL27 (https://www.ebi.ac.uk/chembl/). Each assay has 94 molecular-property pairs on average.

**Training configuration.** For PCQM4Mv1, we set the number of layers as 12 and hidden dimension as 256, which is selected by the cross-validation method on the training set. For FS-Mol, the number of layers are 6 and the hidden dimension is 256. The candidate number of layers and hidden dimensions for MoleculeNet are $\{4, 6, 8, 12\}$ and $\{128, 256\}$. On FS-Mol and MoleculetNet, the hyper-parameters are selected according to validation performance. We train all these tasks on one GPU. The optimizer is AdamW (Loshchilov & Hutter, 2019). More detailed parameters are summarized in Table 5 of Appendix A.

**Results on PCQM4Mv1** The results of PCQM4Mv1 are reported in Table 1. We compare $\mathcal{O}$-GNN with the following baselines: (1) Conventional GCN/GIN with/without virtual node (marked by "vn"). The results are from Hu et al. (2021); (2) ConfDSS (Liu et al., 2021), which predicts quantum properties conditioned on low-cost conformer sets; (3) Two-branch Transformer (Xia et al., 2021), which has a regression head and a classification head that learn from each other; (4) Graphormer (Ying et al., 2021; Shi et al., 2022), the champion solution of PCQM4Mv1. Since

| Method | MAE ($\downarrow$) |
|---|---|
| GCN | 0.1684 |
| GCN + vn | 0.1510 |
| GIN | 0.1536 |
| GIN + vn | 0.1396 |
| ConfDSS | 0.1278 |
| Two-branch Transformer | 0.1237 |
| Graphormer$_{base}$ | 0.1193 |
| Graphormer$_{large}$ | 0.1231 |
| $\mathcal{O}$-GNN (ours) | 0.1148 |

Table 1: Validation MAE on PCQM4Mv1.

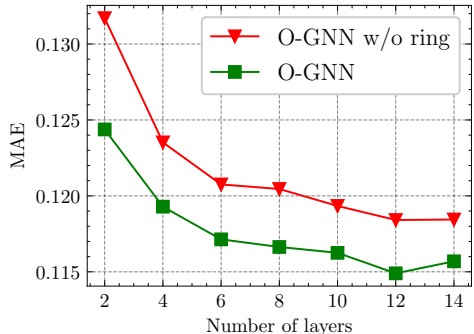

Figure 4: MAE w.r.t. numbers of layers.

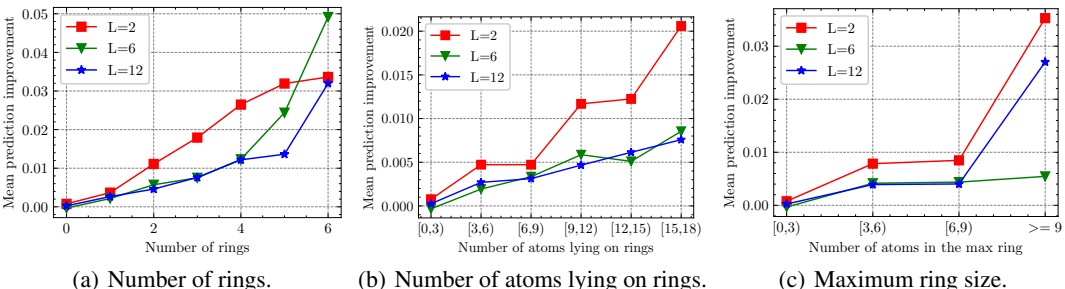

(a) Number of rings.  (b) Number of atoms lying on rings.  (c) Maximum ring size.

Figure 5: Performance improvement over several ring properties.

the owners of PCQM4Mv1 did not release labels of the test set, we can only compare the results on the validation set. The evaluation metric is the mean absolute error (MAE). From Table 1, we can see that $\mathcal{O}$-GNN achieves the best results among the strong baseline models, which shows the effectiveness of our method. In addition, GIN vn, ConfDSS and Graphormer do not explicitly use the ring information, and we will combine $\mathcal{O}$-GNN with the strong methods in the future.

To investigate the significance of the ring information, we study a variant of $\mathcal{O}$-GNN by removing the ring modeling component from $\mathcal{O}$-GNN, and denote this variant as "$\mathcal{O}$-GNN w/o ring". Specifically, it is implemented by removing Eqn.(4) and all the $h_r$'s in Eqn.(2,3,5). We conduct experiments for $\mathcal{O}$-GNN and "$\mathcal{O}$-GNN w/o ring" from 2 to 14 layers. The results are in Figure 4. We can see that by utilizing ring information, the performance is boosted regardless of the number of layers. In addition, we find that a 6-layer $\mathcal{O}$-GNN is comparable with the 12-layer $\mathcal{O}$-GNN w/o ring, which shows the great power of modeling rings in GNN. We also have that $\mathcal{O}$-GNN outperforms "$\mathcal{O}$-GNN w/o ring" in terms of the number of parameters (see Figure 10). It is noteworthy to point out that validation MAE of the 14-layer $\mathcal{O}$-GNN slightly drops compared to the 12-layer $\mathcal{O}$-GNN. Note that this phenomenon is also observed in Graphormer (Shi et al., 2022) that larger models do not always lead to better validation results. We will explore how to train deeper models in the future.

On PCQM4Mv1, we also study the average performance improvement w.r.t. several ring properties. The performance improvement is defined as $\epsilon_1 - \epsilon_2$, where $\epsilon_1$ and $\epsilon_2$ denote the validation MAE of "$\mathcal{O}$-GNN w/o rings" and $\mathcal{O}$-GNN. The ring properties include: (i) the number of rings in a molecule; (ii) the number of atoms lying on rings; (iii) the number of atoms in the largest ring. We conduct experiments for the networks with different numbers of layers ($L = 2, 6, 12$). Results are reported in Figure 5. We can conclude that overall, as the increase of number of rings, maximum ring sizes and the number of atoms lying on rings, $\mathcal{O}$-GNN achieves more improvement compared to the variant without modeling rings. More analyses are in Appendix C.4.

**Results on MoleculeNet** For MoleculeNet, we compare with both pretraining and non-pretraining methods. For non-pretraining methods, we compare with the following baselines: (i) GCN (Kipf & Welling, 2017) with virtual node ; (ii) GIN (Xu et al., 2018) with virtual node; (iii) $\mathcal{O}$-GNN without using ring information (denoted as "$\mathcal{O}$-GNN w/o ring"). For pre-training methods, we select

| Dataset
# Molecules | BBBP
2039 | Tox21
7831 | ClinTox
1478 | HIV
41127 | BACE
1513 | SIDER
1478 |
|---|---|---|---|---|---|---|
| (Hu et al., 2020) | $71.2 \pm 0.9$ | $74.2 \pm 0.8$ | $73.7 \pm 4.0$ | $75.8 \pm 1.1$ | $78.6 \pm 1.4$ | $60.4 \pm 0.6$ |
| G-Contextual (Liu et al., 2022) | $70.3 \pm 1.6$ | $75.2 \pm 0.3$ | $59.9 \pm 8.2$ | $75.9 \pm 0.9$ | $79.2 \pm 0.3$ | $58.4 \pm 0.6$ |
| G-Motif (Liu et al., 2022) | $66.4 \pm 3.4$ | $73.2 \pm 0.8$ | $77.8 \pm 2.0$ | $73.8 \pm 1.4$ | $73.4 \pm 4.0$ | $60.6 \pm 1.1$ |
| GraphMVP (Liu et al., 2022) | $72.4 \pm 1.6$ | $75.9 \pm 0.5$ | $79.1 \pm 2.8$ | $77.0 \pm 1.2$ | $81.2 \pm 0.9$ | $63.9 \pm 1.2$ |
| GCN + vn | $72.7 \pm 1.3$ | $75.0 \pm 0.4$ | $92.0 \pm 1.1$ | $78.8 \pm 1.1$ | $80.0 \pm 0.8$ | $62.9 \pm 1.3$ |
| GIN + vn | $71.7 \pm 0.6$ | $74.8 \pm 0.6$ | $89.4 \pm 3.2$ | $79.3 \pm 1.0$ | $82.0 \pm 1.0$ | $60.8 \pm 0.8$ |
| $\mathcal{O}$-GNN w/o ring | $74.5 \pm 1.4$ | $75.2 \pm 0.9$ | $90.2 \pm 2.1$ | $80.5 \pm 1.0$ | $84.2 \pm 1.5$ | $65.5 \pm 1.6$ |
| $\mathcal{O}$-GNN (ours) | $76.4 \pm 0.4$ | $75.7 \pm 0.7$ | $94.3 \pm 1.6$ | $81.3 \pm 1.2$ | $85.8 \pm 1.0$ | $66.2 \pm 1.2$ |

Table 2: Test ROC-AUC (%) performance of different methods on 6 binary classification tasks from MoleculeNet benchmark. The training, validation and test sets are provided by DeepChem. Each experiment is independently run for three times. The mean and standard derivation are reported.

several representative graph-based methods: (i) Hu et al. (2020) proposed to predict the masked attributes on graphs as well as maintaining the consistency between a subgraph and its neighbors; (ii) G-{Contextual, Motif} are variants of (Rong et al., 2020), which are provided in Liu et al. (2022). (iii) GraphMVP (Liu et al., 2022), which is a joint pre-training between 2D molecules and its 3D conformation. The results of (Hu et al., 2020), G-{Contextual, Motif} and GraphMVP are all extracted from Liu et al. (2022), since Liu et al. (2022) use the same scaffold-based splitting as us.

The results are reported in Table 2. We can see that: (i) $\mathcal{O}$-GNN outperforms the conventional network architectures like GIN and GCN with virtual nodes, which demonstrate the effectiveness of our new architecture; (ii) When comparing with G-{Contextual, Motif}, GraphMVP (Liu et al., 2022) and Hu et al. (2020) which are all pre-training methods, $\mathcal{O}$-GNN still outperforms those methods. (More discussion about pre-training methods are left in Table 11 of Appendix C.5) This shows the great potential of $\mathcal{O}$-GNN and we will combine it with pre-training in the future. (iii) Comparing $\mathcal{O}$-GNN and $\mathcal{O}$-GNN w/o ring, the average improvement overall the six tasks is 1.6. This shows the advantage of using ring information in molecular property prediction.

**Results on FS-Mol** Stanley et al. (2021) verify that prototypical networks (PN) performs the best on FS-Mol compared with other methods like MAML (Finn et al., 2017), multi-task learning (MT) and random forest (RF). Stanley et al. (2021) use a Transformer-like residual network for few-shot classification. We replace that backbone to our $\mathcal{O}$-GNN and "$\mathcal{O}$-GNN w/o ring", and the other parts remain unchanged. Following Stanley et al. (2021), the results with different support set sizes (denoted as $|\mathcal{T}_{u,support}|$) are reported. A support set consists of a few examples with input-label pairs used to train models. The evaluation metric is $\Delta$AUPRC, which is the difference between the AUPRC (area under the precision-recall curve) and the ratio of the active compounds in that query set. A higher $\Delta$AUPRC score indicates better classification performance of the model.

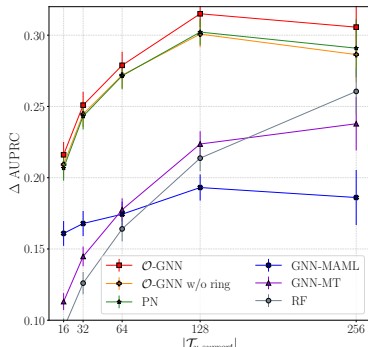

Figure 6: Results on FS-Mol.

The results are in Figure 6. We report the mean and the standard derivations for different tasks across various support sizes. We have the following observations: (i) By using $\mathcal{O}$-GNN as the backbone model for the prototypical network, the results are boosted for different support set sizes. (ii) The improvement is more significant when the support set size is large. When $|\mathcal{T}_{u,support}| = 128$ and 256, the improvements are 0.014 and 0.016. When reducing the sizes to 16/32/64, their improvements are all around 0.008. We will further improve the results on limited data size in the future.

## 3.2 APPLICATION TO DDI PREDICTION

Drug-drug interaction (DDI) prediction is to predict therapeutic output of taking two drugs together, like increasing the risk of some side effects, or the effect is enhanced to take them together. We

focus on the classification task, where the inputs are two drug molecules and one interaction (e.g., inhibition), and the output is 0 or 1 to indicate whether the two drugs have this specific interaction. Following Nyamabo et al. (2022) and Li et al. (2022), we work on the inductive setting of the DrugBank dataset (Wishart et al., 2018), which has $1,706$ drugs, 86 interaction types, and $191,808$ triplets. To test the generalization ability of the model, we conduct experiments on two settings w.r.t. the drugs: the S1 setting, where neither of the two drugs on the test set appears in the training set; the S2 setting, where one drug is seen in the training set and the other is not. Noting that the drug pairs in the test set do not appear in the training set. Hence, the DrugBank data is split into training and test sets by the visibility of the drugs, and the negative samples are offline generated. We directly use the data provided by Nyamabo et al. (2021; 2022), where $20\%$ drugs are first hold as unseen drugs for formulating test set and the rest $80\%$ drugs are used to create the training set.

| Method | S2 setting (1 known drug + 1 unknown drug) | | | | S1 setting (2 unknown drugs) | | | |
| --- | --- | --- | --- | --- | --- | --- | --- | --- |
| | ACC | AUROC | AP | F1 | ACC | AUROC | AP | F1 |
| GAT-DDI (Nyamabo et al., 2021) | 69.83 | 77.29 | 75.79 | 73.01 | 62.63 | 70.92 | 73.01 | 45.81 |
| MHCADDI (Deac et al., 2019) | 70.58 | 77.84 | 76.16 | 72.74 | 65.40 | 73.43 | 75.03 | 54.12 |
| MR-GNN (Xu et al., 2019) | 74.67 | 83.15 | 83.81 | 69.88 | 66.50 | 72.53 | 71.06 | 67.21 |
| SSI-DDI (Nyamabo et al., 2021) | 76.38 | 84.23 | 84.94 | 73.54 | 66.31 | 72.75 | 71.61 | 68.68 |
| GMPNN (Nyamabo et al., 2022) | 77.72 | 84.84 | 84.87 | 78.29 | 68.57 | 74.96 | 75.44 | 65.32 |
| MSAN-GCN (Zhu et al., 2022) | 77.81 | 85.74 | – | 76.48 | 69.17 | 76.12 | – | 67.10 |
| MSN-DDI (Li et al., 2022) | 81.92 | 91.01 | 91.09 | 80.18 | 73.42 | 81.79 | 81.82 | 70.34 |
| $\mathcal{O}$-GNN w/o ring | 87.72 | 94.51 | 95.28 | 85.91 | 75.47 | 83.83 | 85.58 | 65.59 |
| $\mathcal{O}$-GNN (ours) | 88.47 | 95.87 | 96.51 | 86.91 | 76.81 | 87.64 | 88.70 | 70.81 |

Table 3: Results of drug-drug interaction prediction on DrugBank.

To predict the interaction between two drugs, we use one 6-layer $\mathcal{O}$-GNN to extract features for the two drugs. Specifically, for each drug, we average the node representations output by the last layer as the drug feature. We concatenate the two drug features together, and then multiply the interaction embedding to do the prediction. The detailed parameters are left in Table 6 of Appendix A.

The results are reported in Table 3. $\mathcal{O}$-GNN significantly outperforms previous baselines in terms of accuracy (denoted as ACC), the area under the receiver operating characteristic (AUROC), the average precision (AP), and the F1 score. Most of previous works use GCN, GIN or GAT backbones, and they focus on designing comprehensive interaction module (Nyamabo et al., 2021; Li et al., 2022). By using the advanced $\mathcal{O}$-GNN backbone, we can significantly improve the results without designing complex interaction modules. This shows the effectiveness of our method.

### 3.3 APPLICATION TO RETROSYNTHESIS

Retrosynthesis is to predict the reactants of a given product. Various GNNs have been applied to this task. For example, GLN (Dai et al., 2019) use GNNs to predict the distributions of candidate reaction templates and reactants. GraphRetro (Somnath et al., 2021) and G2G (Shi et al., 2020) use GNNs to predict where to break a bond and how to add the fragments to complete the synthons. To demonstrate the ability of our $\mathcal{O}$-GNN, we combine our method with LocalRetro (Chen & Jung, 2021), the current best graph-based model for retrosynthesis (without using pre-training). LocalRetro uses GNN to predict the possible templates for each atom and each bond, and sort the predicted templates according to their probabilities. The top templates will be applied to the corresponding atoms or bonds via `RDKit` (Landrum et al., 2016) to generate the reactants. Chen & Jung (2021) use MPNN (Gilmer et al., 2017a) for prediction, and we replace the MPNN with $\mathcal{O}$-GNN. We conduct experiments on the USPTO-$50k$ dataset (Coley et al., 2017) that contains $50,016$ reactions. Following Chen & Jung (2021), we partition the dataset as $45k$ training set, $5k$ validation set and $5k$ test set. The evaluation metric is the the top-$k$ accuracy, where $k = 1, 3, 5, 10, 50$. The results are summarized in Table 4. We can observe that $\mathcal{O}$-GNN can predict reactions more accurately than the baselines without ring information. Especially, when the reaction type is known, we improve the top-1 accuracy for 1.8 points and the top-3 accuracy for 1.6 points. These results show the importance of modeling ring structure and the effectiveness of our method.

**The performance for different number of rings.** To study the prediction performance of molecules with different number of rings, we group the USPTO-$50k$ test set by the number of rings in the product molecules and compute the top-1 accuracy for each group. More specifically, we divide

| Method | Reaction type unknown | | | | | Reaction type known | | | | |
|---|---|---|---|---|---|---|---|---|---|---|
| | **Top-1** | **Top-3** | **Top-5** | **Top-10** | **Top-50** | **Top-1** | **Top-3** | **Top-5** | **Top-10** | **Top-50** |
| G2G | 48.9 | 67.6 | 72.5 | 75.5 | – | 61.0 | 81.3 | 86.0 | 88.7 | – |
| GLN | 52.5 | 69.0 | 75.6 | 83.7 | 92.4 | 64.2 | 79.1 | 85.2 | 90.0 | 93.2 |
| GraphRetro | 53.7 | 68.3 | 72.2 | 75.5 | – | 63.9 | 81.5 | 85.2 | 88.1 | – |
| LocalRetro | 53.4 | 77.5 | 85.9 | 92.4 | 97.7 | 63.9 | 86.1 | 92.4 | 96.3 | 97.9 |
| $\mathcal{O}$-GNN (ours) | 54.1 | 77.7 | 86.0 | 92.5 | 98.2 | 65.7 | 87.7 | 93.4 | 96.9 | 98.3 |

Table 4: Results on USPTO-$50k$ datasets with reaction type known/unknown.

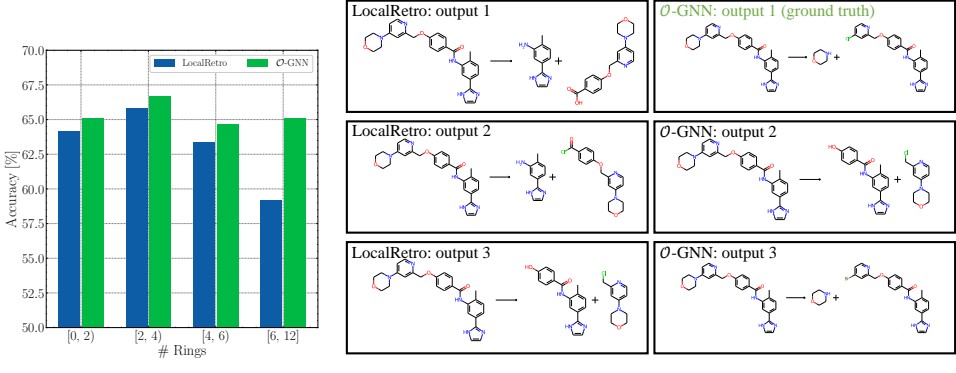

(a) Accuracy w.r.t #rings.    (b) Case study on molecule with complex rings.

Figure 7: Study of $\mathcal{O}$-GNN on retrosynthesis task. (a) The top-1 accuracy w.r.t. number of rings in product molecules. (b) The one-step retrosynthesis prediction of a product molecule with five rings. The first $\mathcal{O}$-GNN output is the same as the ground truth (marked as green).

the test set into four groups with ring numbers $[0, 2)$, $[2, 4)$, $[4, 6)$, $[6, 12]$, and those groups have 808, 2347, 1617, 235 reactions, respectively. The results are in Figure 7(a) where the blue bars represent the LocalRetro baseline and the green bars represent $\mathcal{O}$-GNN. The results show that $\mathcal{O}$-GNN has better accuracy on all groups, where the improvements are 0.99, 0.85, 1.30 and 5.96. Overall speaking, the improvement is larger when there are more rings in a molecule. Especially, when there are at least 6 rings in a group (i.e., the last column), $\mathcal{O}$-GNN increases the accuracy for 5.96 points, demonstrating that our method can better leverage ring structures.

**Case study.** In Figure 7(b), we show an example prediction of a product molecule with 5 rings. The reactions in the left panel are the top-3 predictions from LocalRetro baseline and the ones on the right are from $\mathcal{O}$-GNN. Our method successfully predicts the correct reactants in its first output (marked as green), but the baseline fails to give a correct prediction. More importantly, the baseline system even fails to identify the correct bond to change. These results suggest that modeling ring structures is crucial to predict reactions accurately, and $\mathcal{O}$-GNN is an effective algorithm for retrosynthesis.

## 4    CONCLUSIONS AND FUTURE WORK

In this work, we propose a new model, ring-enhanced GNN (briefly, $\mathcal{O}$-GNN) for molecular modeling. We explicitly incorporate the ring representations into GNN and jointly update them with atom and bond representations. We provide theoretical analysis to $\mathcal{O}$-GNN and prove that by using $\mathcal{O}$-GNN, the node representations are more distinguishable than the variant without using ring representations. We conduct experiments on molecular property prediction, drug-drug interaction (DDI) prediction and retrosynthesis. $\mathcal{O}$-GNN outperforms strong baselines on these tasks and achieves state-of-the-art results on the validation performance of PCQM4Mv1 and DDI prediction. For future work, first, we will combine with pre-training to obtain a stronger $\mathcal{O}$-GNN. Second, we need to further improve our model when the training data is very limited (e.g., when the support set size is 16 or fewer). Third, how to efficiently identify and incorporate the representations with more complex structures is another interesting direction to explore. Fourth, we will apply our model to more real world scenarios, like the synthesis and generation of natural products with large rings.

## ACKNOWLEDGMENTS

This work was supported in part by NSFC under Contract 61836011, and in part by the Fundamental Research Funds for the Central Universities under contract WK3490000007.

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

## A    DETAILED EXPERIMENT CONFIGURATIONS

The hyperparameters for the molecular property prediction, drug-drug interaction prediction and retrosynthesis are summarized in Table 5, Table 6 and Table 7 respectively.

| | PCQM4Mv1 | FS-Mol | MoleculeNet |
|---|---|---|---|
| Number of Layers | 12 | 6 | $\{4, 6, 8, 12\}$ |
| Hidden dimension | 256 | 256 | $\{128, 256\}$ |
| Optimizer | AdamW | AdamW | AdamW |
| Dropout | 0.0 | $\{0.0, 0.1, 0.2\}$ | $\{0.0, 0.1, 0.2, 0.3, 0.5\}$ |
| Learning rate | 0.0003 | $\{0.0001, 0.0002, 0.0003\}$ | $\{0.00005, 0.0001, 0.0002, 0.0005\}$ |
| Training steps | 300 epochs | 10000 iterations | $\{50,100\}$ epochs |
| Batch size | 512 | 16 | $\{32,64\}$ |
| Weight Decay | 0.1 | $\{0.01, 0.1\}$ | 0.01 |
| Learning Rate Decay | Cosine | Cosine | Linear |

Table 5: Detailed hyper-parameters for molecular property prediction tasks.

| Number of Layers | 6 |
|---|---|
| Hidden dimension | 512 |
| Optimizer | AdamW |
| Dropout | $\{0.2, 0.5\}$ |
| Learning rate | 0.0003 |
| Training steps | 100 epochs |
| Batch size | 128 |
| Weight Decay | 0.01 |
| Learning Rate Decay | Cosine |

Table 6: Detailed hyper-parameters for drug-drug interaction prediction.

| Number of Layers | 6 |
|---|---|
| Hidden dimension | 512 |
| Optimizer | AdamW |
| Dropout | 0.1 |
| Learning rate | 0.0003 |
| Training steps | 200 epochs |
| Batch size | 64 |
| Weight Decay | 0.1 |
| Learning Rate Decay | Cosine |

Table 7: Detailed hyper-parameters for retrosynthesis.

## B  PROOFS OF THE TWO PROPOSITIONS

*Proof of Proposition 1.* We start the proof by explicitly writing down the ring-free variant of $\mathcal{O}$-GNN.

Specifically, the bond representations are given by

$$h_{ij}^{(l)} = h_{ij}^{(l-1)} + \texttt{MLP}(h_i^{(l-1)}, h_j^{(l-1)}, h_{ij}^{(l-1)}, U^{(l-1)}).$$

The atom representations are given by

$$\bar{h}_i^{(l)} = \sum_{j \in \mathcal{N}(i)} \alpha_j W_v \texttt{concat}(h_{ij}^{(l)}, h_j^{(l-1)});$$

$$\alpha_j \propto \exp(\boldsymbol{a}^\top \texttt{LeakyReLU}(W_q h_i^{(l-1)} + W_k \texttt{concat}(h_j^{(l-1)}, h_{ij}^{(l)})));$$

$$h_i^{(l)} = h_i^{(l-1)} + \texttt{MLP}\left(h_i^{(l-1)}, \bar{h}_i^{(l)}, U^{(l-1)}\right). \tag{6}$$

The compound representations are given by

$$U^{(l)} = U^{(l-1)} + \texttt{MLP}\left(\frac{1}{|V|} \sum_{i=1}^{|V|} h_i^{(l)}, \frac{1}{|E|} \sum_{i,j} h_{ij}^{(l)}, U^{(l-1)}\right). \tag{7}$$

Given the above notations, Proposition 1 can then be translated to the following claim:

**Claim.** For any valued graphs $(G, f)$ and any two nodes $v_a, v_b$ in $G$, if the valued $k$-neighbourhood sub-graph of $u$ and $v$ (i.e., $(G(v_a, k), f)$ and $(G(v_b, k), f)$) are equivalent, we have that $h_a^l = h_b^l$ for any $l \in \{1, \cdots, k\}$ holds regardless of the parameter.

We denote the equivalent mapping between $(G(v_a, k), f)$ and $(G(v_b, k), f)$ as $\mathcal{P}$. We will slightly abuse the notations by letting $\mathcal{P}(i) = j$ if $\mathcal{P}(v_i) = v_j$.

We will prove the above claim by induction. Specifically, we will prove that for any $l \in \{0, 1, \cdots, k\}$, we have $h_{c_1}^l = h_{\mathcal{P}(c_1)}^l$ for any $v_{c_1} \in V(v_a, k - l)$, and $h_{c_1 c_2}^l = h_{\mathcal{P}(c_1)\mathcal{P}(c_2)}^l$ for any $v_{c_1}, v_{c_2} \in V(v_a, k - l)$.

Base case: for $l = 0$, by the definition of $f$, we have $f(v_{c_1}) = f(\mathcal{P}(v_{c_1}))$ for every $v_{c_1} \in V(v_a, k)$ and $f(v_{c_1}, v_{c_2}) = f(\mathcal{P}(v_{c_1}), \mathcal{P}(v_{c_2}))$ for every $v_{c_1}, v_{c_2} \in V(v_a, k)$. The claim immediately follows as $h_{c_1}^0 = f(v_{c_1})$, $h_{\mathcal{P}(c_1)}^0 = f(\mathcal{P}(v_{c_1}))$, $h_{c_1 c_2}^0 = f(v_{c_1}, v_{c_2})$, and $h_{\mathcal{P}(c_1)\mathcal{P}(c_2)}^0 = f(\mathcal{P}(c_1), \mathcal{P}(c_2))$.

Induction step: suppose the claim is true for $l = i \in \{0, \cdots, k-1\}$. Then, for $l = i+1$, we have that for every $v_{c_1}, v_{c_2} \in V(v_a, k-l)$,

$$
\begin{aligned}
h_{c_1 c_2}^{(l)} =& h_{c_1 c_2}^{(l-1)} + \mathtt{MLP}(h_{c_1}^{(l-1)}, h_{c_2}^{(l-1)}, h_{c_1 c_2}^{(l-1)}, U^{(l-1)}) \\
\overset{(\star)}{=}& h_{\mathcal{P}(c_1)\mathcal{P}(c_2)}^{(l-1)} + \mathtt{MLP}(h_{\mathcal{P}(c_1)}^{(l-1)}, h_{\mathcal{P}(c_2)}^{(l-1)}, h_{\mathcal{P}(c_1)\mathcal{P}(c_2)}^{(l-1)}, U^{(l-1)}) \\
=& h_{\mathcal{P}(c_1)\mathcal{P}(c_2)}^{(l)},
\end{aligned}
\tag{8}
$$

where $Eq.(\star)$ is due to the induction hypothesis, as $v_{c_1}, v_{c_2} \in V(v_a, k-l) \subset V(v_a, k-(l-1))$.

Similarly, for every $v_{c_1} \in V(v_a, k-l)$,

$$
\begin{aligned}
\bar{h}_{c_1}^{(l)} =& \sum_{j \in \mathcal{N}(c_1)} \alpha_j W_v \mathtt{concat}(h_{c_1 j}^{(l)}, h_j^{(l-1)}) \\
\overset{(\circ)}{=}& \sum_{j \in \mathcal{N}(c_1)} \alpha_j W_v \mathtt{concat}(h_{\mathcal{P}(c_1)\mathcal{P}(j)}^{(l)}, h_{\mathcal{P}(j)}^{(l-1)}) \\
\overset{(\diamond)}{=}& \sum_{j \in \mathcal{N}(c_1)} \alpha_{\mathcal{P}(j)} W_v \mathtt{concat}(h_{\mathcal{P}(c_1)\mathcal{P}(j)}^{(l)}, h_{\mathcal{P}(j)}^{(l-1)}) \\
=& \sum_{j \in \mathcal{N}(c_1)} \alpha_j W_v \mathtt{concat}(h_{\mathcal{P}(c_1)j}^{(l)}, h_j^{(l-1)}) \\
=& \bar{h}_{\mathcal{P}(c_1)}^{(l)},
\end{aligned}
$$

where Eq. $(\circ)$ is due to the induction hypothesis and Eq. (8). Eq. $(\diamond)$ is due to

$$
\begin{aligned}
\alpha_j \propto& \exp(\boldsymbol{a}^\top \mathtt{LeakyReLU}(W_q h_{c_1}^{(l-1)} + W_k \mathtt{concat}(h_j^{(l-1)}, h_{c_1 j}^{(l)}))) \\
=& \exp(\boldsymbol{a}^\top \mathtt{LeakyReLU}(W_q h_{\mathcal{P}(c_1)}^{(l-1)} + W_k \mathtt{concat}(h_{\mathcal{P}(j)}^{(l-1)}, h_{\mathcal{P}(c_1)\mathcal{P}(j)}^{(l)}))),
\end{aligned}
$$

and thus $\alpha_j = \alpha_{\mathcal{P}(j)}$ for any $j \in \mathcal{N}(c_1)$.

We then have

$$
\begin{aligned}
h_{c_1}^{(l)} =& h_{c_1}^{(l-1)} + \mathtt{MLP}\left(h_{c_1}^{(l-1)}, \bar{h}_{c_1}^{(l)}, U^{(l-1)}\right) \\
=& h_{\mathcal{P}(c_1)}^{(l-1)} + \mathtt{MLP}\left(h_{\mathcal{P}(c_1)}^{(l-1)}, \bar{h}_{\mathcal{P}(c_1)}^{(l)}, U^{(l-1)}\right) \\
=& h_{\mathcal{P}(c_1)}^{(l)}.
\end{aligned}
$$

Thus, the claim holds for $l = i+1$, and the proof for the induction claim completes. Thus, the claim is true for every $l \in \{0, \cdots, k\}$.

For every $l \in \{0, \cdots, k\}$ $u \in G(v_a, 0) \subset G(v_a, k-l)$. Therefore, we have $h_a^{(l)} = h_{\mathcal{P}(a)}^{(l)} = h_b^{(l)}$, and the proof is completed. $\qquad \square$

*Proof of Proposition 2.* For two equivalent valued sub-graph $(G(v_a, k), f)$ and $(G(v_b, k), f)$, if $v_a$ and $v_b$ lie on different rings, we have

$$
\frac{1}{|R(v_a)|} \sum_{r \in R(v_a)} h_r^{(0)} \neq \frac{1}{|R(v_b)|} \sum_{r \in R(v_b)} h_r^{(0)}.
$$

Therefore, there exists a choice of `MLP`, such that

$$
\begin{aligned}
h_a^{(1)} &= h_a^{(0)} + \mathtt{MLP}\Big(h_a^{(0)}, \bar{h}_a^{(1)}, \frac{1}{|R(v_a)|} \sum_{r \in R(v_a)} h_r^{(0)}, U^{(0)}\Big) \\
&\neq h_b^{(0)} + \mathtt{MLP}\Big(h_b^{(0)}, \bar{h}_b^{(1)}, \frac{1}{|R(v_b)|} \sum_{r \in R(v_b)} h_r^{(0)}, U^{(0)}\Big) \\
&= h_b^{(1)}.
\end{aligned}
$$

The proof is completed. $\qquad\qquad\qquad\qquad\qquad\qquad\qquad\qquad\qquad\qquad\qquad\qquad$ $\square$

## C  MORE ABLATION STUDY

### C.1  NODE REPRESENTATION POOLING V.S. COMPOUND REPRESENTATIONS

We explore the difference between using average pooling $h_\mathcal{G} = \frac{1}{|V|} \sum_{i=1}^{|V|} h_i^{(L)}$ and the compound representation $U^{(L)}$ for classification. We try two networks with different numbers of layers ($L = 6$ and 12). We conduct experiments on PCQM4Mv1 dataset. The validation mean absolute errors (MAE) are reported in Table 8. We can see that using average node pooling is better than using compound representation. This is consistent with the discovery of using virtual node in GIN (Hu et al., 2021). A virtual node can be regarded as a compound representation, which connects to all nodes in graph. When using virtual nodes, it is a common practice to use the average or sum pooling of node representations to represent a graph. One can refer to `https://github.com/ snap-stanford/ogb/blob/1c875697fdb20ab452b2c11cf8bfa2c0e88b5ad3/ examples/lsc/pcqm4m/gnn.py#L60` for the detailed implementation.

|                          | $L = 6$ | $L = 12$ |
| ------------------------ | ------- | -------- |
| Average node pooling     | 0.1171  | 0.1149   |
| Compound representation  | 0.1196  | 0.1167   |

Table 8: Comparison between using average node representations *VS* compound representations.

### C.2  ATTENTIVELY AGGREGATE THE INFORMATION FROM RINGS

In Eqn.(4), we concatenate the sum pooling of atom representations, the sum pooling of bond representations and compound representations to update ring representations. An alternative solution is to use attention models to aggregate the atom and bond representations. We study a variant which updates the ring representations as follows:

$$
h_r^{(l)} = h_r^{(l-1)} + \mathtt{MLP}\Big(h_r^{(l-1)}, \sum_{v_i \in V(r)} \alpha_i^{(l)} h_i^{(l)}, \sum_{e_{ij} \in E(r)} \beta_{ij}^{(l)} h_{ij}^{(l)}, U^{(l-1)}\Big), \tag{9}
$$

In Eqn.(9),

$$
\alpha_i^{(l)} \propto \exp\Big(W_{q1} h_r^{(l-1)} + W_{k1} h_i^{(l)}\Big) \text{ and } \beta_{ij}^{(l)} \propto \exp\Big(W_{q2} h_r^{(l-1)} + W_{k2} h_{ij}^{(l)}\Big), \tag{10}
$$

where the four $W$'s are parameters to be learned. The results are reported in Table 9. We can see that although our method is simple, it can effectively leverage the ring information, and outperform this attention-based variant.

### C.3  $\mathcal{O}$-GNN WITH BRICS

The ring representation used in our method could be considered as a special motif. One might wonder whether other types of new motifs would be helpful. To see the effect, we use BRICS model (Degen et al., 2008) to decompose molecules into fragments. BRICS designs 16 rules to break bonds that can match a set of chemical reactions. The ring representations in Eqn.(2,3,4,5) are replaced by

| | $L = 6$ | $L = 12$ |
|---|---|---|
| $\mathcal{O}$-GNN | 0.1171 | 0.1149 |
| $\mathcal{O}$-GNN with attention models when updating ring representations | 0.1179 | 0.1160 |

Table 9: Comparison between our method and using attention models when updating ring representations.

| | $L = 2$ | $L = 4$ | $L = 6$ | $L = 8$ | $L = 12$ |
|---|---|---|---|---|---|
| $\mathcal{O}$-GNN | 0.1247 | 0.1201 | 0.1181 | 0.1172 | 0.1155 |
| $\mathcal{O}$-GNN w/o rings | 0.1325 | 0.1243 | 0.1222 | 0.1221 | 0.1204 |
| BRICS | 0.1294 | 0.1239 | 0.1219 | 0.1208 | 0.1193 |

Table 10: Comparison between using simple rings (i.e., our method) and using BRICS-based fragments.

these motif representations. The remaining parts remain unchanged. We conduct the experiments on PCQM4Mv1 dataset, and the results are shown in Table 10. Due to time and computation resource limitation, all the models are trained for 200 epochs.

From Table 10, we can conclude that: (1) using simple ring representations achieves better results than using BRICS; (2) in general, using BRICS is better than the variant without using any ring or ring-based fragmentation information. We will keep exploring more segmentation methods.

### C.4    More comparison between $\mathcal{O}$-GNN and $\mathcal{O}$-GNN w/o rings

As a complementary to the analysis of the MAE w.r.t. the number of rings in molecules in Figure 5, we also report the predicted error (i.e., mean absolute error, MAE) of $\mathcal{O}$-GNN and the variant "$\mathcal{O}$-GNN w/o rings" in Figure 8. We can observe that when molecules have no rings, the two methods perform similar. As the number of rings increases from 1 to 6, the MAE increases, and $\mathcal{O}$-GNN always outperforms the "$\mathcal{O}$-GNN w/o rings" variant.

### C.5    Additional discussions

*About over-smoothing* One might be curious that since we build a 12-layer network, whether it suffers from over-smoothing. Actually, Cong et al. (2021) point that "*over-smoothing does not necessarily happen in practice, a deeper model is provably expressive, can converge to global optimum with linear convergence rate, and achieve very high training accuracy as long as properly trained.*" (The words are from (Cong et al., 2021) for accurate expression). In addition, Li et al. (2020) and Addanki et al. (2021) both successfully trained 50+ layer networks. Our method follows

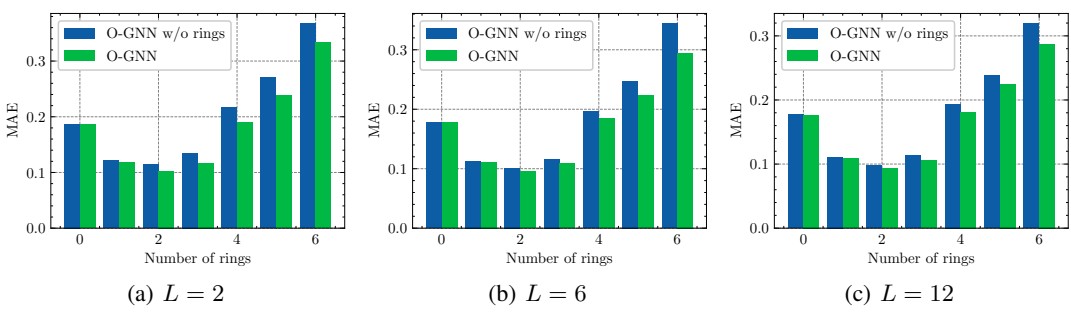

(a) $L = 2$       (b) $L = 6$       (c) $L = 12$

Figure 8: Predicted MAE categorized by different properties. $x$-axis denotes the number or rings, and $y$-axis denoted the mean absolute error (MAE) on the validation set.

the architecture of (Addanki et al., 2021), therefore we do not think that our model suffers from over-smoothing.

*Modeling $k$-neighborhood*: If we want to explicitly use the $k$-neighborhood information, we might need additional modules to process them, like

$$\texttt{net}_1(1 \text{ neighbor nodes}) + \texttt{net}_2(2 \text{ neighbor nodes}) + \cdots + \texttt{net}_k(\text{k neighbor nodes}). \quad (11)$$

To ensure expressiveness, we usually do not share parameters. Therefore, the parameters are $k$ times larger than regular GNN. $\mathcal{O}$-GNN constantly increases the percentages of parameters (irrelevant to $k$). When $k$ is large, $\mathcal{O}$-GNN will be much more parameter efficient. On the other hand, the optimal $k^*$ is not easy to determine. For example, in DrugBank, the ring maximum ring sizes range from 3 (e.g., DB00658) to 53 (e.g., DB05034). Which $k$ is the best is hard to determine.

*About invariant constraints* In $\mathcal{O}$-GNN, the features of atoms, bonds and rings are all invariant. Specifically, the features of atoms and bonds are about their types, number of correlated electrons, number of neighbors, etc (please refer to `https://github.com/O-GNN/O-GNN/blob/5b70a4f9dc9a5f87a0171eea1e9cecde30489eb8/ogb/utils/features.py#L2` for details). The ring representations are obtained via atom and bond representations (please kindly refer to Eqn.(4)), which are also invariant. The variant features (like coordinates) are not encoded.

*Comparison about the convergence speed*: The validation MAE curves of PCQM4Mv1 are shown in Figure 9. The results of 6-layer $\mathcal{O}$-GNN (with/without rings) 12-layer $\mathcal{O}$-GNN (with/without rings) are reported. We can see that:

(1) by training the 6-layer $\mathcal{O}$-GNN for 175 epochs, the results are almost the same as training the 12-layer "$\mathcal{O}$-GNN w/o ring" for 275 epochs;

(2) by training the 12-layer $\mathcal{O}$-GNN for 75 epochs, the results are almost the same as training the 12-layer "$\mathcal{O}$-GNN w/o ring" for 275 epochs.

These results demonstrate that $\mathcal{O}$-GNN has better convergence speed.

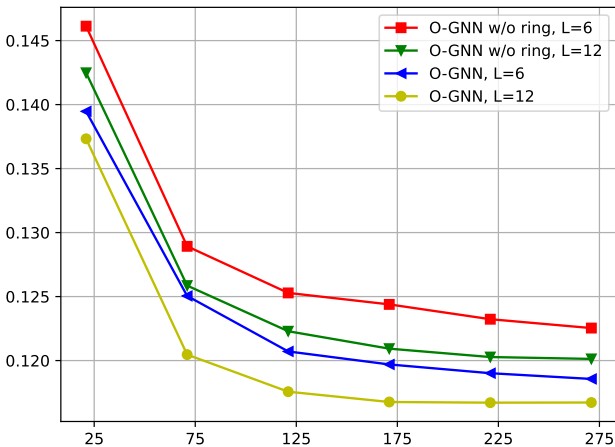

Figure 9: Comparison about the convergence speed of $\mathcal{O}$-GNN and "$\mathcal{O}$-GNN w/o ring". $x$-axis denotes the training epoch and $y$-axis denotes the validation MAE.

*Comparison between different number of parameters.* Figure 4 shows the validation MAE of $\mathcal{O}$-GNN and "$\mathcal{O}$-GNN w/o ring" w.r.t. the number of layers. We also visualize the validation MAE w.r.t. the number of parameters in Figure 10. We can observe that when aligned with the number of parameters, $\mathcal{O}$-GNN still outperforms the variant without modeling rings.

*Pre-training baselines on MoleculeNet.* We summarize the pre-training baselines on MoleculeNet in Table 11. Sun et al. (2022) have demonstrated different data splitting method could result in significantly different results. We follow the common practice to use scaffold based splitting, and we cite the results of Rong et al. (2020) from Fang et al. (2022). Note that the results of $\mathcal{O}$-GNN is not pre-trained on unlabeled molecules. We can see that in terms of the average score, our method

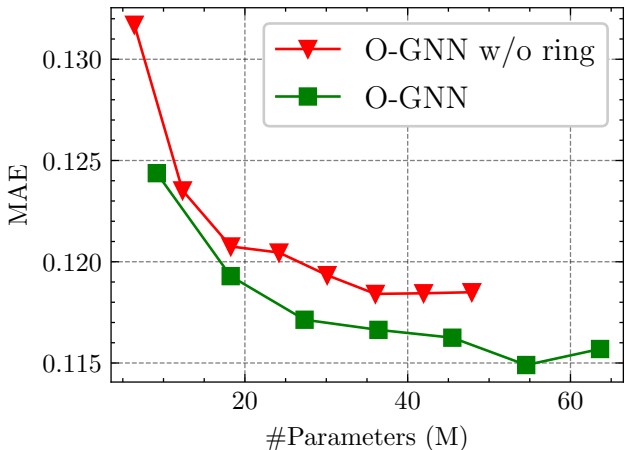

Figure 10: Validation MAE of PCQM4Mv1 w.r.t. the number of parameters.

is comparable with those strong baselines, which demonstrate the effectiveness of our method. We will combine our method with pre-training in the future.

| Dataset
# Molecules | BBBP
2039 | Tox21
7831 | ClinTox
1478 | HIV
41127 | BACE
1513 | SIDER
1478 | Avg |
|---|---|---|---|---|---|---|---|
| (Hu et al., 2020) | $71.2 \pm 0.9$ | $74.2 \pm 0.8$ | $73.7 \pm 4.0$ | $75.8 \pm 1.1$ | $78.6 \pm 1.4$ | $60.4 \pm 0.6$ | 72.3 |
| G-Contextual (Liu et al., 2022) | $70.3 \pm 1.6$ | $75.2 \pm 0.3$ | $59.9 \pm 8.2$ | $75.9 \pm 0.9$ | $79.2 \pm 0.3$ | $58.4 \pm 0.6$ | 69.8 |
| G-Motif (Liu et al., 2022) | $66.4 \pm 3.4$ | $73.2 \pm 0.8$ | $77.8 \pm 2.0$ | $73.8 \pm 1.4$ | $73.4 \pm 4.0$ | $60.6 \pm 1.1$ | 70.9 |
| GraphMVP (Liu et al., 2022) | $72.4 \pm 1.6$ | $75.9 \pm 0.5$ | $79.1 \pm 2.8$ | $77.0 \pm 1.2$ | $81.2 \pm 0.9$ | $63.9 \pm 1.2$ | 74.9 |
| MGSSL (Zhang et al., 2021) | $70.5 \pm 1.1$ | $76.5 \pm 0.3$ | $80.7 \pm 2.1$ | $79.5 \pm 1.1$ | $79.7 \pm 0.8$ | $61.8 \pm 0.8$ | 74.8 |
| GROVER$_{base}$ (Rong et al., 2020) | $70.0 \pm 0.1$ | $74.3 \pm 0.1$ | $81.2 \pm 3.0$ | $62.5 \pm 0.9$ | $82.6 \pm 0.7$ | $64.8 \pm 0.6$ | 72.6 |
| GROVER$_{large}$ (Rong et al., 2020) | $69.5 \pm 0.1$ | $73.5 \pm 0.1$ | $76.2 \pm 3.7$ | $68.2 \pm 1.1$ | $81.0 \pm 1.4$ | $65.4 \pm 0.1$ | 72.3 |
| GEM (Fang et al., 2022) | $72.4 \pm 0.4$ | $78.1 \pm 0.1$ | $90.1 \pm 1.3$ | $80.6 \pm 0.9$ | $85.6 \pm 1.1$ | $67.2 \pm 0.4$ | 79.0 |
| $\mathcal{O}$-GNN (ours) | $76.4 \pm 0.4$ | $75.7 \pm 0.7$ | $94.3 \pm 1.6$ | $81.3 \pm 1.2$ | $85.8 \pm 1.0$ | $66.2 \pm 1.2$ | 80.0 |

Table 11: Pre-training baselines on MoleculeNet.

## D RELATED WORK SUMMARY

GCN (Kipf & Welling, 2017) aggregates its neighbor information according to the adjacency matrix and degree matrix, and then updates the aggregated information with a linear transformation and a non-linear activation layer. GraphSAGE (Hamilton et al., 2017b) aggregates the neighbors information by element-wise average. GAT (Veličković et al., 2017) introduces the attention mechanism into GNN, by which it can adaptively aggregates the representations of the neighbors. Brody et al. (2021) propose the GATv2 to improve the attention mechanism in a more expressive way. Xu et al. (2018) develop a simple aggregate function which involves an $\epsilon$ parameter and multi-layer perceptrons (MLPs) which is provably as powerful as the Weisfeiler-Lehman graph isomorphism test. Besides, Gilmer et al. (2017b); Li et al. (2017); Pham et al. (2017) propose to augment the graph with a virtual node to capture the global information of the graph. A virtual node connects to all the other nodes in the graph and is jointly updated during training. Its effectiveness is validated in a series of graph classification tasks.

However, these work do not explicitly use the ring $R$ in graph neural networks. Complementary to these work, we consider how to incorporate the ring information, which is another important component on top of the node and edge information, into molecular modeling. These advanced AGGREGATE and UPDATE functions are also applicable to our work.

