# OpenReview forum: "$\mathcal{O}$-GNN: incorporating ring priors into molecular modeling"
_ICLR.cc/2023/Conference — ICLR 2023 poster_

### Official Review · Reviewer_4qh9 · 2022-10-23

**Confidence:** 4
**Correctness:** 4
**Technical Novelty And Significance:** 3
**Empirical Novelty And Significance:** 2
**Recommendation:** 6

**Clarity, Quality, Novelty And Reproducibility:**

The paper is clearly written and presents a novel idea (as far as I can judge). I think in terms of quality, it's lacking due to the evaluation issues I mention above. Code is not provided, yet the model description is clear and straightforward and one could reimplement it.

**Strength And Weaknesses:**

(+) I think it is worth to study molecular ring encoding in an ICLR paper since it concerns an important part of molecules, an important target domain, and representation learning.

(+) The proposed inclusion as first-order components into the MPNN makes totally sense to me and is nice since simple and straightforward.

(-) From an ML perspective, I don't think, the topic and model are investigated fairly. In this form, I would consider the paper to target a more general audience and it could be presented, e.g., at a chemistry venue. More specifically, I am missing a more detailed evaluation and comparison how it improves upon the regular GNN. For example:
- How do the samples look like (in terms of numbers and size of rings) on which the proposed model is particularly better than other models. Are there samples with rings which the other models get right and how do they look like?
- Molecular rings tend to not as many atoms and we have technology to construct deeper GNNs. The question is how and if the proposed model improves over those as well (e.g., what happens after 12 layers in Figure 3).
- Rings can be considered as motifs so it would be interesting to include some GNNs considering motifs into the evaluation.

Note that there is a small section "The performance for different number of rings.", but I think much more such analysis should be given.

================================================================================

Minor Comments

- Sec 3.2 (theoretical analysis) seems unnecessary to me. It's a lot of definitions and reading where, in my opinion, you don't learn much in the end given that it's rather obvious that encoding the rings directly adds expressivity.

- p. 3 "We initlaize"

- p. 2: (I think)
"BACKGROUND AND PRELIMINARY" -> "PRELIMINARIES"
"Notations:" -> "Notation:"

- below Eq (1), the entire paragraph about GNNs is redundant for this paper I would say.



**Summary Of The Paper:**

The paper proposes a new graph neural network O-GNN for reasoning over molecular graphs that specifically encodes ring structures and claims and shows that this improves upon regular GNNs in terms of number of layers required. This is achieved by extending regular message passing GNNs by updating  ring encodings (using simple feed-forward networks) along with the usual update of node and edge encodings.


**Summary Of The Review:**

Altogether, I think the authors have a good idea but do not justify it in terms of investigation and evaluation for an ML venue focusing on representation learning. A detailed analysis would probably make the paper rather different and go beyond minor corrections. I tend to rejection at the current point. I am open to discussion since I think the paper makes an interesting contribution.

---

> ### Author Response · Authors · 2022-11-14
> **Response to Reviewer 4qh9**
>
> Thanks for your valuable review comments!
> > **[Q1] How do the samples look like (in terms of numbers and size of rings) on which the proposed model is particularly better than other models.**
>
> **A**: We add analyses on PCQM4Mv1, where we study the performance improvement w.r.t the following properties: (1) the number of rings in molecules; (2) the number of atoms lying on in rings; (3) the size of maximum rings. The performance improvement is defined as $\epsilon_1-\epsilon_2$, where $\epsilon_1$ is the validation MAE of “O-GNN w/o rings” and $\epsilon_2$ denotes the validation MAE of our O-GNN. The improvement becomes larger as the increase of the above three properties.  Please kindly refer to Figure 5 of our revised paper and the paragraph above "Results on MoleculeNet".
>
> > **[Q2] Are there samples with rings which the other models get right and how do they look like?**
>
> **A**: Indeed, there are molecules  that regular GNN (e.g., O-GNN w/o rings) get better MAE than O-GNN. Currently, we do not find a clear pattern of the molecules that regular GNN can handle while our method cannot. We will keep exploring them.
>
>
> > **[Q3] what happens after 12 layers in Figure 4? (Figure 4 corresponds to the Figure 3 in the initial submission)**
>
> **A**:  If we keep increasing the number of layers to 14, the validation performance of O-GNN drops slightly (see Figure 4), but still better than previous baselines. For the Graphormer baseline in Table 1, we also observe that larger models do not always lead to better validation performance. We will explore how to further improve the performances as we increase the number of layers. The discussion is added in “Results on PCQM4Mv1”
>
>
>
> > **[Q4] Rings can be considered as motifs so it would be interesting to include some GNNs considering motifs into the evaluation.**
>
> **A**: Following your suggestion, we use BRICS to segment molecules into motifs, and embed it into our network.  BRICS designs 16 rules to break bonds that can match a set of chemical reactions.  The ring representations in Eqn. (2,3,4,5) are replaced by these motif representations. The remaining parts remain unchanged. We conduct the experiments on PCQM4Mv1 dataset, and results are shown as follows:
>
> || L=2| L=4|L=6|L=8|L=12|
> | -- | -- | -- |-- | -- | -- |
> |O-GNN | 0.1247|0.1201|0.1181|0.1172|0.1155|
> |O-GNN w/o ring | 0.1325|0.1243|0.1222|0.1221|0.1204|
> |BRICS|0.1249|0.1239|0.1219|0.1208|0.1193|
>
>
> From the above table, we can conclude that: (1) using simple ring representations achieves better results than using BRICS; (2) in general, using BRICS is better than the variant without using any ring or ring-based fragmentation information. We will keep exploring more segmentation methods.
>
> The revision can be found at Appendix C.3.
>
>
> > **[Q5] Reply to Minor Comments**
>
> **A**: Thanks for pointing the typos. We also move the entire paragraph below Eqn.(1) to Appendix D.

---

> > ### Comment · Reviewer_4qh9 · 2022-11-16
> > **Thank you for the response.**
> >
> > All the details really clarified the picture. I am leaning towards acceptance. The only reason against which I see is the fact that, from an ML-perspective, the novelty of the architecture is limited. But I think that's a more general aspect of how such application papers are to be treated.

---

> > > ### Author Response · Authors · 2022-11-17
> > > **Response to Reviewer 4qh9**
> > >
> > > **Thanks a lot for your positive and valuable comments!**
> > >
> > > About the novelty of our work, we would like to clarify the following points:
> > > 1. The rings are important components for molecular modeling, but are not explicitly leveraged in prior work. We design a simple yet effective network to leverage the ring information.
> > > 2. The experimental results on a series tasks, including molecular property prediction, retrosynthesis and drug-drug interaction, have demonstrated the effectiveness of our method.
> > > 3. We hope our method  could inspire more network architectures to incorporate the chemical priors in the future.
> > >
> > > Thanks again!

---

### Official Review · Reviewer_Q5iY · 2022-10-25

**Confidence:** 4
**Correctness:** 3
**Technical Novelty And Significance:** 2
**Empirical Novelty And Significance:** 2
**Recommendation:** 5

**Clarity, Quality, Novelty And Reproducibility:**

The presentation of this paper is clear, and code is provided for reproduction. However, the novelty is limited.


**Strength And Weaknesses:**

Strength:
1. \mathcal{O}-GNN achieves better performance than baseline methods for both molecular property prediction and retrosynthesis prediction tasks.
2. It is interesting to model rings separately in the molecules since rings play an important role in molecular properties.

Weaknesses:
1. The innovation of the model is not good enough. It simply updates the ring representations by concatenating atom and bond representations.
2. The theoretical analysis is not convincing. In fact the ring representations are obtained by aggregating the representations of multi-hop nodes on the ring. A similar discriminatory ability can be achieved by sampling multiple nodes in the k-neighborhood to update the central node.
3. The proposed method is used for molecular modeling but ignores the equivariant constraints among atoms, bonds and rings.



**Summary Of The Paper:**

In this paper, the authors propose a ring-enhanced GNN called \mathcal{O}-GNN to learn ring representations for molecular modeling. The ring representations are updated by concatenating the atom representations and bond representations in the ring, as well as the overall compound representation. Experiments on 11 public datasets present good performance of \mathcal{O}-GNN.


**Summary Of The Review:**

In this paper, the authors propose to improve the discriminative ability of GNN in the field of molecular modeling by updating the ring representations. Although the proposed model performs better than the baseline approach on several public available datasets, there is still space for improvement in model innovation. In addition to the weaknesses mentioned above, I have the following concerns:

1. I'm curious how the proposed method overcomes the oversmoothing problem by stacking so many layers?
2. Why not consider more other types of subgraphs, such as hinge.
3. Some operations should be supported by adding references or experiments.
* Why is it necessary to concatenate compound representations when updating atom, bond and ring representations?
* Why not use the learned compound representations for graph classification tasks but use the average pooling results of atomic representations?
4. In Table 1, why does graphormer-large perform worse than graphormer-base?
5. In Table 2，is there any explanation as to why wo ring's O-GNN works better than baseline?

---

> ### Author Response · Authors · 2022-11-14
> **Response to Reviewer Q5iY [Part 1]**
>
> Thanks for your valuable review comments!
> > **[Q1] The innovation of the model is not good enough. It simply updates the ring representations by concatenating atom and bond representations.**
>
> **A**: We would like to clarify that:
> - What we want to propose is a simple yet effective method that can leverage the ring information of molecules. Rings are important components of molecules but not properly modeled in previous work. Therefore, we propose the method in this paper. According to the results on PCQM4M, MoleculeNet, FS-Mol and DDI, if we remove the ring representations, the performance drops significantly.
>
> - We also study a variant to update the ring representations. Instead of using the sum pooling of atom/bond representations, we use an attention model to obtain the ring representation. The results are not as good as the current plan. The content is latex-heavy, and you could kindly go to Appendix C.2 to check the results.
>
>
> > **[Q2] The theoretical analysis is not convincing. In fact the ring representations are obtained by aggregating the representations of multi-hop nodes on the ring. A similar discriminatory ability can be achieved by sampling multiple nodes in the k-neighborhood to update the central node.**
>
> **A**: Thanks for your question. However, we respectively argue that modeling rings still play a unique role than “sampling multiple nodes in the k-neighborhood to update the central node”.
>
> If we want to explicitly use the $k$-neighborhood information in one layer, we might need additional modules to process them, like $f_1(1\~neighbor\~nodes) + f_2(2\~neighbor\~nodes) + … + f_k(k\~neighbor\~nodes)$. To ensure expressiveness, we usually do not share parameters. Therefore, the parameters are $k$ times larger than regular GNN. In comparison, O-GNN constantly increases the percentages of parameters (irrelevant to $k$). When $k$ is large, O-GNN will be much more parameter efficient.
>
> The optimal $k^*$ is not easy to determine. For example, in DrugBank, the maximum ring sizes range from 3 (e.g., DB00658) to 53 (e.g., DB05034). It is hard to select a unified $k$, and using a dynamic $k$ also requires additional efforts.
>
> We add the above discussion in Appendix C.5 .
>
>
>
> > **[Q3] The proposed method is used for molecular modeling but ignores the equivariant constraints among atoms, bonds and rings.**
>
> **A**: In O-GNN, the features of atoms, bonds and rings are all invariant. Specifically, the features of atoms and bonds are about their types, number of correlated electrons, number of neighbors, etc (please refer to [url](https://github.com/O-GNN/O-GNN/blob/5b70a4f9dc9a5f87a0171eea1e9cecde30489eb8/ogb/utils/features.py#L2) for details). The ring representations are obtained via atom and bond representations (please kindly refer to Eqn.(4)), which are also invariant. The variant features (like coordinates) are not encoded. The current O-GNN processes the 2D molecules. When dealing with 3D conformations in the future, we will also pay attention to the equivariance/invariance.
>
> We add the above discussion in Appendix C.5 .
>
>
>
> > **[Q4] About novelty**
>
> **A**: Rings are important components of molecules. However, they are less explored in GNN literature. We introduce them into GNN. What we want to show the audience is a simple yet effective way to leverage this prior.
>
> We propose a new architecture that models the ring information, and achieves state-of-the-art performances on PCQM4M, FS-MOL, DDI and retrosynthesis.
>
>
>
> > **[Q5] About oversmoothing**
>
> **A**: Cong. et al (2021) pointed that “we argue that over-smoothing does not necessarily happen in practice, a deeper model is provably expressive, can converge to global optimum with linear convergence rate, and achieve very high trainin accuracy as long as properly trained.” (words from the [ref1]). Both [ref2] and [ref3] have trained GNN with more than 50 layers respectively. We follow the architecture designed by [ref3], therefore we do not think that our model suffers from over-smoothing.
>
> We add the above discussion to Appendix C.5.
>
>
> [ref1] On Provable Benefits of Depth in Training Graph Convolutional Networks, https://proceedings.neurips.cc/paper/2021/file/524265e8b942930fbbe8a5d979d29205-Paper.pdf
>
> [ref2] DeeperGCN: All You Need to Train Deeper GCNs, https://arxiv.org/pdf/2006.07739.pdf
>
> [ref3] Large-scale graph representation learning with very deep GNNs and  self-supervision, Ravichandra Addanki et al, https://arxiv.org/abs/2107.09422

---

> > ### Author Response · Authors · 2022-11-14
> > **Response to Reviewer Q5iY [Part 2]**
> >
> > > **[Q6] Why not consider more other types of subgraphs, such as hinge.**
> >
> > **A**: Sorry but we do not know what you mean by “hinge”. Can you provide a reference? If you are talking about the “hinge regions” in proteins [https://www.sciencedirect.com/topics/medicine-and-dentistry/hinge-region] or immunoglobulin [https://pubmed.ncbi.nlm.nih.gov/828971/],  thanks for providing us the reference but our current focus is small molecules, not proteins.
> >
> > Moreover, we conduct another experiment by using BRICS to segment molecules into motifs, and embedding it into our network. BRICS designs 16 rules to break bonds that can match a set of chemical reactions. The ring representations in Eqn. (2,3,4,5) are replaced by these motif representations. The remaining parts remain unchanged. We conduct the experiments on PCQM4Mv1 dataset, and results are shown as follows:
> >
> >
> >
> > || L=2| L=4|L=6|L=8|L=12|
> > | -- | -- | -- |-- | -- | -- |
> > |O-GNN | 0.1247|0.1201|0.1181|0.1172|0.1155|
> > |O-GNN w/o ring | 0.1325|0.1243|0.1222|0.1221|0.1204|
> > |BRICS|0.1249|0.1239|0.1219|0.1208|0.1193|
> >
> >
> > From the above table, we can conclude that: (1) using simple ring representations achieves better results than using BRICS; (2) in general, using BRICS is better than the variant without using any ring or ring-based fragmentation information. We will keep exploring more segmentation methods.
> >
> > The revision can be found at Appendix C.3.
> >
> >
> >
> >
> > > **[Q7] Why is it necessary to concatenate compound representations when updating atom, bond and ring representations?**
> >
> > **A**: We follow the architecture of [ref4, ref5, ref6], which concatenates the compound representations when updating atom and bond representations. The compound representation in our method is similar to the virtual node in these work, which connects to all nodes and captures the global representations. When updating the node representations, virtual node representations are still used. This is consistent with our design.
> >
> > [ref4] Neural Message Passing for Quantum Chemistry, https://arxiv.org/pdf/1704.01212.pdf
> >
> > [ref5] Learning Graph-Level Representation for Drug Discovery, https://arxiv.org/pdf/1709.03741.pdf
> >
> > [ref6] Graph Classification via Deep Learning with Virtual Nodes, https://arxiv.org/pdf/1708.04357.pdf
> >
> >
> >
> > > **[Q8] Why not use the learned compound representations for graph classification tasks but use the average pooling results of atomic representations?**
> >
> > **A**: The main inspiration is from GIN-virtual. Even if with a virtual node, it is still recommended to use the pooling of all nodes. You could refer to [url](https://github.com/snap-stanford/ogb/blob/1c875697fdb20ab452b2c11cf8bfa2c0e88b5ad3/examples/lsc/pcqm4m/gnn.py#L60) for the example. Therefore, we follow this pattern.
> >
> > Following your suggestion, we explore the difference between using averaging pooling $h_{\mathcal{G}}=\frac{1}{\vert V\vert}\sum_{i=1}^{\vert V\vert}h_i^{(L)}$ and the compound representation $U^{(L)}$ for classification. We try two networks with different numbers of layers ($L=6$ and $12$). We conduct experiments on PCQM4Mv1 dataset. The mean absolute errors (MAE) are shown as follow:
> >
> > || L=6| L=12 |
> > | -- | -- | -- |
> > Average pooling| 0.1171| 0.1149       |
> > Compound representation| 0.1196   | 0.1167        |
> >
> >  We can see that using average node pooling is better than using compound representation.  These results are added in Appendix C.1.
> >
> >
> > > **[Q9] In Table 1, why does graphormer-large perform worse than graphormer-base?**
> >
> > **A**: The results are copied from their original paper. As stated in their paper, the authors claim that “Interestingly, although the optimization error is lower, the generalization ability of Graphormer-Large is worse on this dataset”. That is, Graphormer-large is overfitting and thus has higher validation MAE.
> >
> >
> >
> > > **[Q10] In Table 2, is there any explanation as to why wo ring's O-GNN works better than baseline?**
> >
> > **A**: The O-GNN w/o rings variant is similar to [ref7], with the following two main differences: (1) We use attention model to aggregate the node representation, while [ref7] simply uses sum pooling. (2) We add batch normalization layers (empirically better than layer normalization). Therefore, O-GNN w/o rings is a stronger variant of [ref7]. [ref7] achieves very promising results in KDDCup challenge (only 0.0004 gap to the first one and significantly better than GIN or GCN), and it is supposed to be a stronger model on molecular modeling.
> >
> > [ref7] Large-scale graph representation learning with very deep GNNs and self-supervision, Ravichandra Addanki et al, https://arxiv.org/abs/2107.09422

---

> > > ### Author Response · Authors · 2022-11-17
> > > **Looking forward to the feedback**
> > >
> > > Dear Reviewer Q5iY,
> > >
> > > Thank you for your review comments. Considering the deadline of the Discussion Stage 1 is approaching, is there anything we can do for our paper?
> > >
> > > Thanks,
> > >
> > > Authors

---

> > > ### Comment · Reviewer_Q5iY · 2022-11-18
> > > **About hinge**
> > >
> > > [Q6] The hinge here indicates a structure with 3 atoms and 2 bonds linked in line.
> > >
> > > The major contribution of the model is limited since it only consider a special aggregation of one kind of subgraph, ring. Actually, there are many kinds of functional groups in the molecules besides rings. However, no other functional groups are considered in the paper.
> > >
> > > Also, I reviewed the experiments again, the results for GROVER are much worse than the original paper. As shown in Table 2 of [1], the performance of GROVER without pre-train is still better than the proposed method.
> > >
> > > |    |BBBP|SIDER|ClinTox|BACE|Tox21|ToxCast|
> > > |---|--------|---------|----------|--------|--------|-----------|
> > > |GROVER w/o pretain| 0.911|0.624|0.884|0.858|0.803|0.721|
> > > |$\mathcal{O}$-GNN|0.764|0.662|0.943|0.858|0.757| - |
> > >
> > > Since the authors claims that the results for SOTAs come from Liu et. al. 2022, I reviewed that paper as well and find that Liu et. al. 2022 actually replaced the backbone of GROVER from a graph transformer to GIN which actually is a different method from GROVER.

---

> > > > ### Author Response · Authors · 2022-11-18
> > > > **Response**
> > > >
> > > > Thanks for the reply, but we are afraid that you mis-understand our work.
> > > >
> > > > > **Reply to "hinge" and "only consider a special aggregation of one kind of subgraph, ring"**
> > > >
> > > > 1. We have stated the importance of rings in the first paragraph of introduction and Figure 1. You could kindly check them.  In addition, a recent paper re-emphasizes that “The structural core of most small-molecule drugs is formed by a ring system”. [https://pubs.rsc.org/en/content/articlehtml/2022/np/d2np00001f]. Thanks for pointing out the “hinge” but we are not quite sure what "hinge" generally means for small molecules.
> > > >
> > > > 2. Ring systems are the basic component of many function groups. Therefore, we start from ring. We have claimed that in the conclusion section, "Third, how to efficiently identify and incorporate the representations with more complex structures is another interesting direction to explore."
> > > >
> > > > 3. In the rebuttal period, we add a section about using BRICS to segment molecules (please go to Appendix C.3), which can split molecules into segments with not only rings, but also chains. O-GNN still outperforms this variant.
> > > >
> > > > > **Reply to "GROVER"**
> > > >
> > > > 1. The original GROVER paper does not use the same split as the default DeepChem data provider. The original GROVER uses a split called “balanced_scaffold” [https://github.com/tencent-ailab/grover/blob/0421d97a5e1bd1b59d1923e3afd556afbe4ff782/grover/util/utils.py#L449] while more works adopt the purely “scaffold” based data split, like the default DeepChem, our work, [ref1,ref2,ref3], and (Liu et al 2022). [ref4] has demonstrated different data splitting method could result in significantly different results (see the TABLE 2,3,4,5 in [ref4]).
> > > > 2. Thanks for pointing out the backbone problem and we would like to change the name from GROVER to "G-{contextual,motif}" and revise the text accordingly.
> > > > 3. Considering that we do not pre-train O-GNN on unlabeled molecules, in Table 2, we list several pre-training methods which use the same data split as references. We also add a discussion about pre-training baselines in Table 11 of Appendix C.5.
> > > >
> > > > [ref1] Geometry-enhanced molecular representation learning for property prediction, NMI, https://www.nature.com/articles/s42256-021-00438-4,
> > > >
> > > > https://github.com/PaddlePaddle/PaddleHelix/blob/dev/apps/pretrained_compound/ChemRL/GEM/scripts/finetune_class.sh#L63
> > > >
> > > > [ref2] Motif-based Graph Self-Supervised Learning for Molecular Property Prediction, NeurIPS, https://proceedings.neurips.cc/paper/2021/file/85267d349a5e647ff0a9edcb5ffd1e02-Paper.pdf
> > > >
> > > > [ref3] Strategies for pre-training graph neural networks, ICLR, https://openreview.net/pdf?id=HJlWWJSFDH
> > > >
> > > > [ref4] Does GNN Pretraining Help Molecular Representation? NeurIPS, https://openreview.net/pdf?id=uytgM9N0vlR

---

> > > > > ### Comment · Reviewer_Q5iY · 2022-11-21
> > > > > **Thank you for your response**
> > > > >
> > > > > > Other functional groups:
> > > > >
> > > > > I see. According to the results in Table 10, it means that the model using only rings may performs better than that using more segment molecules. It is an interesting finding. However, I still think the proposed method is lack of novelty in methodology. Therefore, I will keep a borderline score for this paper.
> > > > >
> > > > > > Inconsistent results in experiments.
> > > > >
> > > > > Thank you for your explanation. I reviewed the new references and agree that the original GROVER only uses balanced scaffold splitting. It is more reasonable to use random scaffold splitting instead.

---

> > > > > > ### Author Response · Authors · 2022-11-21
> > > > > > **Response to  Reviewer Q5iY**
> > > > > >
> > > > > > Thanks for your reply.
> > > > > >
> > > > > > 1. The importance of rings has been illustrated in the first paragraph of introduction and in Figure 1 and Figure 2. Incorporating domain priors into models has always been attractive. What we propose is a simple yet effective way to use ring information.
> > > > > >
> > > > > > 2. Rings vary in terms of sizes, atom types, aromatic types, etc. According to our statistics on $3.8M$ molecules from PCQM4Mv1, there are $17.7K$ unique rings covering $88.06\\%$ molecules in the PCQM4Mv1 dataset. Those rings could be one of the representative factors to distinguish molecules.
> > > > > >
> > > > > > 3. For the comparison with BRICS, it is indeed interesting to see that our method outperforms the BRICS-based variant. How to choose the effective segment algorithms is indeed nontrivial and we will explore more along this direction in the future.

---

### Official Review · Reviewer_X4DK · 2022-10-25

**Confidence:** 3
**Correctness:** 3
**Technical Novelty And Significance:** 3
**Empirical Novelty And Significance:** Not applicable
**Recommendation:** 8

**Clarity, Quality, Novelty And Reproducibility:**

**Clarity**

The work is well structured. However, I cannot say the same about the writing quality. It seems to be written in a hurry, and it is riddled with typos and generally poor wording. I was tempted to signal the typos, but they were so many I had to desist.

**Quality**

I'd say medium, considering that the clarity is low.

**Novelty**

Explicitly modeling rings in molecular GNNs is the main contribution of this paper, which to my knowledge has not been done up until now. So, this work qualifies as novel (unless I'm mistaken, will dig more deeply into the literature before the rebuttal phase begins).

**Reproducibility**

The code is provided in an anonymous repo. Though I haven't replicated the experiments, it's nice to see that the authors were kind enough to make their code available.

**Strength And Weaknesses:**

**Strengths**

- It makes sense from a chemical point of view to have a separate modeling "circuit" for rings in molecules.
- Really impressive results across different tasks.

**Weaknesses**

- Not really well-written (although well-organized).
- I am a bit confused by some of the experimental results. The PCQM4Mv1 and MolNet benchmarks are somewhat okay. On FS-Mol, $\mathcal{O}$-GNN replaces a backbone transformer-based residual network. However, the results using "$\mathcal{O}$-GNN without rings" as a backbone are not shown, so it is not clear if the improvement is due to the use of GNN or because of the ring modeling. A similar pattern happens in the drug-drug interaction prediction task.
- The theoretical analysis shows that few layers of $\mathcal{O}$-GNN are at least as expressive as many more layers of a regular-GNN. But this does not necessarily qualify as an advantage in practice, unless **1)** $\mathcal{O}$-GNN layers uses fewer parameters than a regular GNN with the same expressivity; **2)** $\mathcal{O}$-GNN is faster to train than a regular GNN with the same expressivity. To put it in a different way, why do I have to use few layers of $\mathcal{O}$-GNN, when I can get the same expressivity with many regular-GNN layers (and perhaps also using fewer parameters or being computationally faster)? I wonder if the authors can provide evidence that using $\mathcal{O}$-GNN is really more advantageous than using regular-GNN once the expressivity is similar.

**Summary Of The Paper:**

In the context of predictive tasks which involve molecules, the paper presents a GNN variant called $\mathcal{O}$-GNN that explicitly models rings in compounds, besides the usual atom/bond modeling. The paper presents a theoretical analysis that justifies why it is preferable to account for rings in molecules, as well as a series of experiments to show that the model performs well in practical usage. Notably, this model is able to outperform the winner of the KDDCup on the PCQM4Mv1 benchmark (although on the validation set).

**Summary Of The Review:**

**Disclaimer**

I did not check the proofs.

**Overall judgment**

With the doubts I have exposed above, I consider this paper borderline, and I'm giving a 5 for the moment. I'd really like the author to respond to my points in order to be able to raise my score. Also, I **highly** recommend getting this paper proofread for the rebuttal. It would be a shame if I had to leave my score as-is just because nothing was done to improve the writing.

**Questions/comments to authors**

- I think an image will help the reader understand the "two isomorphic sub-graphs lying on different rings" in Section 1. Also, this statement is a bit misleading since two graphs with different rings are not isomorphic. Do you mean isomorphic in structure, but not in the node/bond features that make up the ring? I think this difference is addressed in Section 3.2, however, reading up to this point, the reader is confused.
- In Eq. 4, why the ring representation at the previous layer $h_r^{(l-1)}$ is inside the $\texttt{MLP}$, when the other representations (Eq. 2, 4, 5) are outside it?
- For the graph classification tasks, why do you do an average pooling to get the graph representation, when you have all the compound representations at each layer already available?


**Minors**
Page 7: define multi-task learning as MT and random forest as RF in the text, so they are readily understandable when looking at Figure 4.

Page 7: although it is clear from context, you should specify that $\Delta$-AUPRC is a metric where higher values are better. You should also explain what a "support set" is.



**EDIT**

After 1st rebuttal, I am changing my score to 6 given the impressive effort put by the authors to answer my enquiries.


**EDIT**

After a successful rebuttal, I am now convinced this paper deserves acceptance, and I'm therefore raising my score to 8.

---

> ### Author Response · Authors · 2022-11-14
> **Response to Reviewer X4DK [Part 1]**
>
> Thanks for your valuable review comments!
> > **[Q1] Concerns about paper writing.**
>
> **A**: Thanks for your detailed review comments. Following your suggestions,, we polish the introduction, fix the typos, add several illustrations for theoretical results and add new results.
>
> > **[Q2] On FS-Mol, O-GNN replaces a backbone transformer-based residual network. However, the results using "O-GNN without rings" as a backbone are not shown, so it is not clear if the improvement is due to the use of GNN or because of the ring modeling. A similar pattern happens in the drug-drug interaction prediction task.**
>
> **A**: We add the baseline of “O-GNN w/o rings” for the FS-MOL (see Figure 6, the orange curve) and DDI (see Table 3).
>
> For the FS-Mol, O-GNN outperforms the “O-GNN w/o rings” variant on different support set sizes. For DDI, our method outperforms the variant on different settings (S2 setting and S1 setting) and different evaluation metrics (accuracy, ROCAUC, AP and F1). This shows that the improvement is from ring modeling.
>
> > **[Q3] The theoretical analysis shows that few layers of O-GNN are at least as expressive as many more layers of a regular-GNN. But this does not necessarily qualify as an advantage in practice, unless 1) O-GNN layers uses fewer parameters than a regular GNN with the same expressivity; 2) O-GNN is faster to train than a regular GNN with the same expressivity. To put it in a different way, why do I have to use few layers of O-GNN, when I can get the same expressivity with many regular-GNN layers (and perhaps also using fewer parameters or being computationally faster)?**
>
> **A**: Thanks for your questions. We provide the following answers.
>
> - Reply to “O-GNN layers uses fewer parameters than a regular GNN with the same expressivity”: Actually, from Proposition 1 and 2, we can get this claim. Without ring representations, the GNN requires at least $k$ layers to distinguish two isomorphic $k$-neighborhood nodes in a ring, while O-GNN only requires one layer for this purpose. We also give an example in Figure 2 of the main paper. Comparing O-GNN and a regular GNN with the same number of layers, modeling ring presentations constantly increases the percentages of parameters (irrelevant to $k$). However, a regular GNN may require $k$ layers to achieve the discriminatory power for $k$-neighborhood sub-graphs. When $k$ is large, O-GNN will be much more parameter efficient.
>
> - Reply to “O-GNN is faster to train than a regular GNN with the same expressivity”: We have not conducted theoretical analysis for this. However, we add some curves between O-GNN and “O-GNN w/o rings” to demonstrate that O-GNN is faster to train. Specifically,
>
>     1.  By training the 6-layer O-GNN for 175 epochs, the results are almost the same as training the 12-layer `` O-GNN w/o ring'' for 275 epochs;
>
>     2.  By training the 12-layer O-GNN for 75 epochs, the results are almost the same as training the 12-layer `` O-GNN w/o ring'' for 275 epochs.
>
>     You could refer to Appendix C.5, paragraph “Comparison about the convergence speed” for the figure and analysis.
>
>
>
> > **[Q4] I think an image will help the reader understand the "two isomorphic sub-graphs lying on different rings" in Section 1.**
>
> **A**: Thanks for your helpful suggestions. We add Figure 2 in the introduction (i.e., Section 1). In this figure, the three substructures in the red circles are isomorphic. The second and third substructures lie on different rings (a Cyclooctane and an Azocane). A regular GNN requires multiple layers to distinguish the three substructures while O-GNN requires only one layer due to the ring representations.
>
> > **[Q5] In Eq. 4, why the ring representation at the previous layer hr(l−1) is inside the MLP, when the other representations (Eq. 2, 4, 5) are outside it?**
>
> **A**: Sorry, this is a typo. It has the representations outside. You can refer to the [url](https://github.com/O-GNN/O-GNN/blob/main/dualgraph/gnn.py#L218).
> We have updated it in Eqn. (4).

---

> > ### Author Response · Authors · 2022-11-14
> > **Response to Reviewer X4DK [Part 2]**
> >
> > > **[Q6] For the graph classification tasks, why do you do an average pooling to get the graph representation, when you have all the compound representations at each layer already available?**
> >
> > **A**: The main inspiration is from GIN-virtual. Even if with a virtual node, it is still recommended to use the pooling of all nodes. You could refer to [url](https://github.com/snap-stanford/ogb/blob/1c875697fdb20ab452b2c11cf8bfa2c0e88b5ad3/examples/lsc/pcqm4m/gnn.py#L60) for the example. Therefore, we follow this pattern.
> >
> > Following your suggestion, we explore the difference between using averaging pooling $h_{\mathcal{G}}=\frac{1}{\vert V\vert}\sum_{i=1}^{\vert V\vert}h_i^{(L)}$ and the compound representation $U^{(L)}$ for classification. We try two networks with different numbers of layers ($L=6$ and $12$). We conduct experiments on PCQM4Mv1 dataset. The mean absolute errors (MAE) are shown as follow:
> >
> > || L=6| L=12 |
> > | -- | -- | -- |
> > Average pooling| 0.1171| 0.1149       |
> > Compound representation| 0.1196   | 0.1167        |
> >
> >  We can see that using average node pooling is better than using compound representation.  These results are added in Appendix C.1.
> >
> >
> >
> > > **[Q7] Page 7: define multi-task learning as MT and random forest as RF in the text, so they are readily understandable when looking at Figure 4.**
> >
> > **A**: Thanks. We have added them. You could refer to the blue words in the paragraph “Results on FS-Mol”

---

> > > ### Comment · Reviewer_X4DK · 2022-11-15
> > > **Thanks for the effort (and some additional clarification)**
> > >
> > > Thank you for your massive effort in answering all my doubts! I am definitely leaning toward acceptance now, and have updated my score accordingly.
> > >
> > > Also thanks for figure 2, it helps a lot. Perhaps in Figure 2A) you should change $HN_2$ to $H_2N$? Otherwise, they are trivially distinguishable. Am I wrong?
> > >
> > > As regards Q3, I still need a bit of clarification, maybe because I wasn't very clear in my question. I will try a simplified example, and please correct me if my reasoning is faulty. Suppose that bonds, atoms, rings, and compounds' hidden states all have a representation of size 16 (and assume no bias weights). Then, the bond, atom, ring, and compound MLPs will have 16 x 16 x 4 x 5=5120 parameters. In contrast, without ring representation, you would have 16 x 16 x 3 x 4=3072 parameters. So without using ring representations you "save" 5120-3072=2048 parameters at each layer. This means that, for example, if you have a 2-layer $\mathcal{O}$-GNN (which has 10240 parameters), a "fairer" comparison would be to use a 3-layer $\mathcal{O}$-GNN w/o rings, since it has 9216 parameters (or a 4-layer which has 12288), and so on. I suggest to produce a Figure similar to Figure 4, where instead of the number of layers in the x-axis you use the number of network parameters.
> > >
> > > Because in the end, I don't care how many layers have been used, I just care that, given the same number of parameters, $\mathcal{O}$-GNN performs better than $\mathcal{O}$-GNN w/o rings. Hope it's clearer now.

---

> > > > ### Author Response · Authors · 2022-11-15
> > > > **Response to additional clarification**
> > > >
> > > > Thanks a lot for your positive and valuable comments!
> > > >
> > > > > **About the error in Figure 2A**
> > > >
> > > >
> > > >
> > > > **A**: Sorry for misleading due to the small font size. Actually, it is $NH_2$ not $HN_2$, and we have adjusted the font size for better understanding.
> > > >
> > > >
> > > >
> > > > > **Figure about network parameters**
> > > >
> > > >
> > > >
> > > > **A**: Following your suggestion, we add Figure 10 in Appendix.  We can observe that when aligned with the number of parameters, O-GNN still outperforms the variant without modeling rings.

---

> > > > > ### Comment · Reviewer_X4DK · 2022-11-17
> > > > > **Thanks!**
> > > > >
> > > > > Thank you! I am now fully satisfied of this rebuttal. I think with these additions this work is worthy of an 8. Wish you luck!

---

> > > > > > ### Author Response · Authors · 2022-11-17
> > > > > > **Thanks!**
> > > > > >
> > > > > > Thank a lot for your efforts on improving our work!

---

### Decision · Program_Chairs · 2023-01-20

**Decision:**

Accept: poster

**Justification For Why Not Higher Score:**

Theoretical analysis are unconvincing and the description are a bit unclear.

**Justification For Why Not Lower Score:**

Adequate technical novelty and experimental results.

**Metareview: Summary, Strengths And Weaknesses:**

The paper proposes a ring-enhanced GNN called O-GNN to learn ring representations for molecular modeling. The ring representations are updated by concatenating the atom representations and bond representations in the ring, as well as the overall compound representation. Experiments on 11 public datasets present good performance of O-GNN.

Strength of paper:
1. the idea of modelling rings separately.
2. Experiments show O-GNN outperforms baseline methods for both molecular property prediction and retrosynthesis prediction tasks.

Weakness of the paper:
1. The theoretical analysis is not convincing.
2. Writing could be improved. Experimental results are a bit unclear.

Overall, the paper proposes an interesting novel idea of molecule representation with experimental support.

**Note From Pc:**

if the above contains the word "oral" or "spotlight" please see: "oral" presentation means -> notable-top-5% and "spotlight" means -> notable-top-25%. As stated in our emails, we are disassociating presentation type from AC recommendations